# UNCERTAINTY-AWARE TREE SEARCH FOR EFFICIENT LLM REASONING

## ABSTRACT

Tree search is an important class of methods for improving the multi-step reasoning capability of Large Language Models (LLMs) by explicitly exploring the intermediate steps on a search tree guided by a value function. However, the existing tree-search methods often devote equal computational budgets across different reasoning branches regardless of the associated uncertainty, causing significantly high token consumption. In this paper, we first conduct a pilot study to reveal the ubiquitous semantic redundancy of reasoning trajectories starting from an intermediate reasoning step. Such highly certain reasoning steps will ultimately reduce the diversity of the final answers. Further, we theoretically show that under a probabilistic guarantee, the sampling budget required to maintain fixed generation quality grows proportionally with the step uncertainty. Built on top of this, we propose **Uncertainty-Aware Allocation** (UAA), a plug-and-play framework that allocates search budget adaptively according to the step-wise uncertainty. In particular, UAA detects for highly certain reasoning steps and incorporates (i) uncertainty-aware pruning (UAP) to keep only a high-quality subset of candidate actions, and (ii) uncertainty-aware budgeting (UAB) to shrink the next-step expansion budget. Extensive empirical evaluations demonstrate that UAA can significantly reduce token consumption and wall-clock time without hurting accuracy when applied to Beam Search and MCTS.

## 1 INTRODUCTION

Large Language Models (LLMs) have demonstrated impressive capabilities in natural language understanding and reasoning (Anil et al., 2023; Chowdhery et al., 2023; Achiam et al., 2023; DeepSeek-AI et al., 2025; Yang et al., 2025; NVIDIA et al., 2025), but they still struggle with complex tasks requiring accurate multi-step reasoning, such as mathematical problem solving (Lightman et al., 2023; Ahn et al., 2024), logical deduction (Wei et al., 2025; Lin et al., 2025), and agent planning (Banerjee et al., 2025). To address this, Chain-of-Thought (CoT) (Wei et al., 2022) guides LLMs to decompose the problem-solving process into multiple intermediate reasoning steps. CoT improves interpretability and performance, but remains vulnerable to errors in each reasoning step, e.g., calculation errors or misinterpretations, that ultimately degrade the quality of the response.

To enhance the reliability of multi-step reasoning, tree-search techniques, including Beam Search (Yao et al., 2023; Xie et al., 2023; Kang et al., 2024; Snell et al., 2024), and Monte Carlo Tree Search (MCTS) (Hao et al., 2023; Feng et al.; Jiang et al., 2024; Wang et al., 2024b), have recently gained attention. These methods improve solution reliability by exploring high-value reasoning steps in a search tree. However, given the vast space of natural language reasoning steps, these methods often incur significant computational overheads (Grosse et al., 2024; Wang et al., 2025a), making them impractical for real-world applications. For example, MCTS often requires a significant number of improvement iterations to balance exploration and exploitation, leading to excessive computational overhead, e.g., consuming over 8,000 tokens on a simple mathematical task (see Section 5). To improve the efficiency of tree-search methods, recent approaches (Wang et al., 2024a; 2025a; Liu et al., 2025) merge similar reasoning steps or prune redundant ones during search. By leveraging step-level similarities, these methods reduce the number of explored steps and thereby reduce the overall token consumption of tree-search methods.

However, the existing acceleration techniques still face significant challenges in computational cost and generality. Specifically, they often require training additional models to estimate step-level similarities, which not only brings extra training and serving overheads but also lacks transferability across different reasoning tasks and datasets. Moreover, most existing approaches focus on merging or pruning redundant actions at the current reasoning step (Tian et al., 2024; Wang et al., 2025a), yet overlook the prevalent redundancy in subsequent reasoning trajectories starting from the current reasoning step (see Appendix B.3). As a result, these methods tend to over-explore semantically saturated regions of the search space, leading to unnecessary computational overhead. These limitations raise a natural question: *Can we design a simple method to accelerate tree-based reasoning by leveraging internal information alone, without external models and compromising performance?*

To understand the relationship between redundant reasoning trajectories and final answer diversity, we first conduct a pilot study on Beam Search. The results show that the reasoning trajectories starting from intermediate steps often exhibit strong semantic similarity and eventually lead to similar final answers. This suggests that exploring highly certain reasoning steps contributes little to the coverage of different answers. Moreover, we provide an information-theoretic analysis to formalize this intuition: under a probabilistic guarantee, the sampling budget required to maintain fixed generation quality grows proportionally with the step uncertainty, implying that allocating the full budget to low-uncertainty actions is redundant and should be reduced. Motivated by these insights, we present Uncertainty-Aware Allocation (UAA), a training-free, plug-and-play framework that dynamically allocates budgets for each reasoning step in the search tree. Instead of training an additional estimator, UAA leverages the cosine similarities as a proxy for uncertainty to detect highly certain reasoning steps. To effectively reduce unnecessary explorations, UAA employs (i) uncertainty-aware pruning (UAP) to keep only a high-quality subset of candidate actions, and (ii) uncertainty-aware budgeting (UAB) to shrink the next-step expansion budget. Specifically, our key contributions are summarized as follows.

1. We empirically analyze reasoning trajectories generated by tree-search methods and find that remaining reasoning trajectories from the same step exhibit high semantic similarity. However, this redundancy does not contribute to answer accuracy and instead results in unnecessary computational overhead.

2. We propose a *training-free*, *plug-and-play* framework, named *Uncertainty-Aware Allocation* (UAA), that lowers computational cost in terms of both tokens and latency, requires no auxiliary models or fine-tuning, and is applicable to both *black-box* and *white-box* LLM settings. UAA could accelerate tree-based reasoning via dynamically pruning redundant reasoning steps and allocating expansion budget according to step-wise uncertainty.

3. We evaluate UAA on three math reasoning benchmarks under both Beam Search and MCTS. Results show that UAA consistently reduces inference cost without compromising accuracy and generalizes well across different LLMs and block-box settings. For example, on GSM8K, UAA reduces the average token usage of reward-guided Beam Search from 3.12k to 2.58k and shortens wall-clock time from 53.32s to 34.23s (See Table 1).

## 2 PRELIMINARIES

LLM-based reasoning can be formalized as a sequence generation problem involving multiple intermediate steps. Specifically, given an input question $q$, the generation process of the final answer $\mathbf{a}$ with LLMs can be decomposed into a sequence of $T$ individual reasoning steps, $\mathbf{a} := [a_1, a_2, \ldots, a_T]$, where $a_t$ denotes the $t$-th reasoning step. In practice, each step $a_t$ can be a single line of text or a fixed number of tokens output by LLMs.

We define the state at step $t$ as $s_t := [q, a_1, \ldots, a_{t-1}]$, representing the concatenation of the input question $q$ and the partial reasoning trace generated up to step $t-1$, with the special case $s_1 = q$. At each step, the next state is updated via $s_{t+1} = [s_t, a_t]$, after the LLM generates $a_t$. Then, we define $\mathcal{A}$ as the action space and $\mathcal{S}$ as the state space. $A_t$ and $S_t$ are the random variables corresponding to $a_t$ and $s_t$. The language model's policy $\pi_\theta$, parameterized by $\theta$, defines a conditional distribution $\pi_\theta(a_t|s_t)$ over the next reasoning step $a_t$ given the current state $s_t$.

Tree search methods (Lightman et al., 2023; Wang et al., 2023; Yao et al., 2023; Hao et al., 2023; Wang et al., 2024b) aim to optimize reasoning trajectories by exploring the space of intermediate

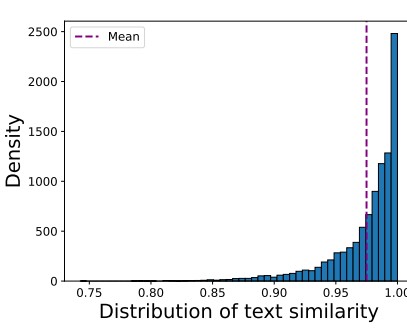 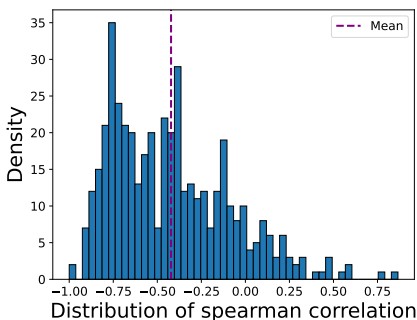

(a) Similarity of reasoning trajectories

(b) Spearman correlation between similarities of reasoning trajectories and diversities of final answers

Figure 1: Motivational analysis of the reasoning tree structures.

reasoning steps guided by a value function. Specifically, the reasoning process can be visualized as a tree, where each node corresponds to a state $s_t$, and each edge represents the action $a_t$ transitioning from state $s_t$ to state $s_{t+1}$. The root node represents the initial state $s_1$, and a path from the root to any leaf node constitutes a complete reasoning trajectory. The goal of tree-search methods is to find the optimal reasoning trajectory $[a_1^*, a_2^*, \ldots, a_T^*]$, given a value function $V$:

$$\mathbf{a}^* = \underset{[a_1, a_2, \ldots, a_T] \in \mathcal{A}}{\arg\max} V(q, a_1, a_2, \ldots, a_T), \tag{1}$$

where $\mathcal{A}$ be the space of reasoning trajectories. The value function $V$ is used to evaluate how well action $a_t$ align with solving the question $q$ given state $s_t$, often denoted as $V(s_t, a_t) \in \mathbb{R}$. Beam search (BS) (Yang et al., 2018; Xie et al., 2023; Feng et al.) is an effective and simple tree search method. Specifically, BS maintains $N$ active beams and expands each by sampling $M$ candidates per step. These candidates are scored by the value function, and the top $N/M$ are retained for the next iteration. This process continues until complete reasoning paths are constructed.

**Value functions** In this work, we consider two types of value functions:

**(i) Model Confidence.** This measures the LLM's belief in the trajectory by averaging the log-probabilities of generated tokens (Yang et al., 2018; Hao et al., 2023; Feng et al.; Snell et al., 2024; Huang et al., 2023b; Fu et al., 2025):

$$V_c(s_t, a_t) = \frac{\sum_{z=1}^{t} \sum_{k=1}^{\ell_{a_z}} \log \pi_\theta(b_{z,k} \mid s_z, b_{z,1}, \ldots, b_{z,k-1})}{\sum_{z=1}^{t} \ell_{a_z}}, \tag{2}$$

where $a_t = [b_{t,1}, \ldots, b_{t,\ell_{a_t}}]$ is the token sequence of step $a_t$. The confidence acts as a prior during search, reflecting the model's preference for the trajectory. While model confidence provides an internal score, it is not always reliable: incorrect reasoning steps can still receive high confidence, potentially leading the search toward suboptimal paths.

**(ii) Process Reward Models (PRMs).** These offer an external, learning-based signal to complement the model's own confidence and enable value-guided search strategies that promote logical consistency and factual correctness throughout the reasoning trajectory (Lightman et al., 2023; Wang et al., 2023).

## 3   PILOT STUDY ON REASONING TRAJECTORY REDUNDANCIES

In this section, we conduct an empirical study on GSM8K (Cobbe et al., 2021) to reveal the ubiquitous redundancies among reasoning trajectories in tree-search methods, and demonstrate that such redundancies would ultimately reduce the diversity of the final answers. Specifically, given a partial reasoning trace $s_t = [q, a_1, \ldots, a_{t-1}]$, we define its continuation, i.e., a sequence of reasoning steps starting from $s_t$, as $p_{s_t}^c = [a_t, \ldots, a_T]$. The redundancy is measured by first independently sampling $M$ continuations $p_{s_t,1}^c, \ldots p_{s_t,M}^c$, and then computing pairwise cosine similarity between the text

embeddings of the continuations, where the text embeddings are extracted from a pre-trained encoder. The redundancy analysis is performed over all intermediate states generated during the search process.

Figure 1 presents the results of reward-guided beam search (Yao et al., 2023; Snell et al., 2024) on *LLaMA3.1-8B-Instruct* (Grattafiori et al., 2024). Here, we consider the beam width of 20, and for each node in the beam we generate 20 child nodes. Full implementation details are deferred to Appendix B. It can be observed that the majority of continuations have a semantic similarity score above 0.95, indicating prevalent redundancy in the reasoning trajectories. Moreover, the mean value of similarity marked by the dashed line is also close to 1, which also confirms the low diversity of continuations starting from a reasoning step $s_t$.

We further explore how the semantic similarity of the reasoning trajectories correlates with the diversity of final answers. To this end, we count the number of unique answers generated by the continuations starting from each reasoning step, and compute the Spearman correlation between the average pairwise cosine similarity and the number of unique answers. It can be seen from Figure 1(b) that the correlation is predominantly negative, with the mean value close to -0.5, which indicates that redundancies among reasoning trajectories would ultimately reduce the diversity of the final answers. In other words, **exploring a highly certain reasoning step whose continuations are semantically similar contributes little to the coverage of different answers**. Therefore, uniform allocation of computational budgets across reasoning steps in a search tree can be highly ineffective, which necessitates an uncertainty-aware allocation policy. More empirical studies on MCTS, Qwen3-32B (Yang et al., 2025), and StrategyQA (Geva et al., 2021) can be found in Appendix C.

## 4 METHOD

In this section, we theoretically analyze the relationship between the current reasoning step and the final answer quality from an uncertainty perspective, showing that allocating small expansion budgets to low-uncertainty steps could reduce computation without materially affecting generation quality. Building on top of this, we present *Uncertainty-Aware Allocation* (UAA), a novel and flexible framework for accelerating existing tree-search algorithms for LLM reasoning.

### 4.1 THEORETICAL ANALYSIS

We begin our analysis by investigating the relationship among the current reasoning step $s_t$, next action $a_t$, and the final answer $y$. Let $Y$ be the random variable for the ground-truth answer $y$. Proposition 1 formalizes the information gain of $A_t$ in the current state $S_t = s_t$ using conditional mutual information.

**Proposition 1** (Ton et al., 2024)**.** *Let* $\mathcal{I}(Y; A_t \mid S_t = s_t)$ *denote the mutual information between random variables $Y$ and $A_t$ conditional on $S_t = s_t$. Then*

$$\mathcal{I}(Y; A_t \mid S_t = s_t) = H(Y \mid S_t = s_t) - H(Y \mid A_t, S_t = s_t) \geq 0. \qquad (3)$$

Since $H(Y \mid S_t = s_t)$ is independent of $A_t$, the information gain of $A_t$ is fully captured by the conditional entropy $H(Y \mid A_t, S_t = s_t)$. We estimate this entropy via Monte Carlo by sampling $n$ candidate actions $\{a_t^{(i)}\}_{i=1}^n \overset{\text{i.i.d.}}{\sim} \pi_\theta(\cdot \mid s_t)$ and using the estimator $\widehat{H}_n := \frac{1}{n} \sum_{i=1}^n h(a_t^{(i)})$, where $h(a_t^{(i)}) := H(Y \mid A_t = a_t^{(i)}, S_t = s_t) = -\sum_y p(y \mid a_t^{(i)}, s_t) \log p(y \mid a_t^{(i)}, s_t)$ denotes the conditional entropy of $Y$ after taking action $a_t^{(i)}$ at state $s_t$, i.e., the residual uncertainty about $Y$.

**Proposition 2.** *Let $Y$ be a target variable, and let $A_t \sim \pi_\theta(\cdot \mid S_t = s_t)$ denote a random action sampled from the policy at state $s_t$. Define the conditional entropy as $H(Y \mid A_t, s_t) := \mathbb{E}_{a_t \sim \pi_\theta}[h(a_t)], \quad where \; h(a_t) := -\sum_y p(y \mid a_t, s_t) \log p(y \mid a_t, s_t)$. Let $\hat{H}_n := \frac{1}{n} \sum_{i=1}^n h(a_t^{(i)})$ be the average of i.i.d. samples $\{a_t^{(i)}\}_{i=1}^n$. If $h(\cdot)$ is $C$-Lipschitz, then for any $\varepsilon > 0$,*

$$\mathbb{P}\left(\left|\hat{H}_n - H(Y \mid A_t, s_t)\right| \geq \varepsilon\right) \leq \frac{C^2 \cdot \text{Var}(A_t \mid s_t)}{n\varepsilon^2}. \qquad (4)$$

*where $\text{Var}(A_t \mid s_t)$ is the variance (i.e., uncertainty ) of $A_t$ conditional on $s_t$.*

The proof of Proposition 2 is provided in Appendix E. Building on Proposition 2, we obtain the following corollary that explicitly links the number of samples with the uncertainty of $t$-th step.

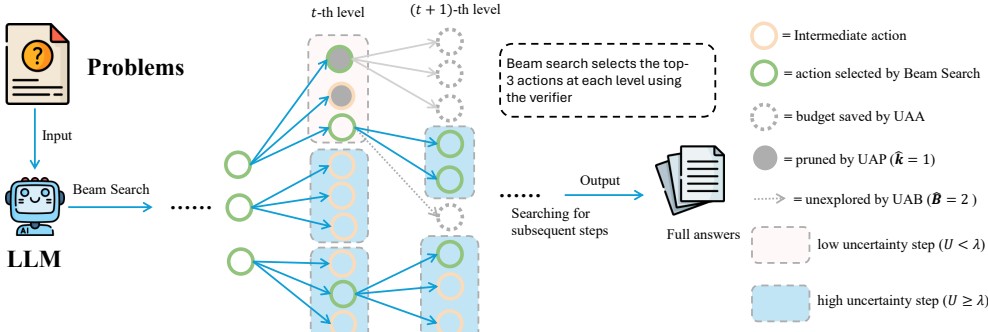

Figure 2: Illustration of UAA integrated with Beam Search. At each reasoning level, the LLM generates candidate actions evaluated by the verifier. For low-uncertainty steps, UAP prunes low-value candidates and UAB reduces the expansion budget, whereas for high-uncertainty steps, full exploration is preserved. "$U$" refers to the uncertainty of sibling actions, formally defined in Equation 6.

**Corollary 1.** *For fixed $\varepsilon$ and $\delta \in \mathbb{R}$ such that $\mathbb{P}\left(\left|\hat{H}_n - H(Y \mid A_t, s_t)\right| \geq \varepsilon\right) \leq \delta$, the required number of samples $n$ grows proportionally with the uncertainty $\mathrm{Var}(A_t \mid s_t)$:*

$$n \geq \frac{C^2 \cdot \mathrm{Var}(A_t \mid s_t)}{\delta \varepsilon^2}.$$

Consequently, according to Corollary 1, in a vanilla tree-search routine, allocating smaller expansion budgets to low-uncertainty steps could reduce computation without materially affecting generation quality. We further validate this in Appendix D: using a fixed token budget, Figure 10 shows that steps whose candidate actions exhibit high redundancy yield negligible marginal answer accuracy gains, supporting Corollary 1.

### 4.2 OUR METHOD: UNCERTAINTY-AWARE ALLOCATION

Driven by the above theoretical analysis, we propose *Uncertainty-Aware Allocation* (UAA), a dynamic budget allocation framework for tree-search algorithms with LLMs, which avoids exploring unnecessary intermediate steps while preserving the overall reasoning quality. The key idea of UAA is to leverage the uncertainty of $A_t$ to determine the subsequent reasoning budget. Formally, given a reasoning step $s_t$ to be expanded, the LLM policy $\pi_\theta(\cdot \mid s_t)$ first proposes up to $B_t$ candidate actions $\mathcal{C}_t = \{a_t^{(1)}, \ldots, a_t^{(B_t)}\} \subseteq \mathcal{A}$. Then UAA reduces unnecessary exploration by (1) selecting a subset $\widehat{\mathcal{C}}_t \subseteq \mathcal{C}_t$ for further expansion according to the uncertainty (pruning), and (2) assigning a suitable expansion budget of eligible candidates $\hat{\mathcal{C}}_t$ (budgeting). Here, the uncertainty $U_t := U(A_t \mid S_t=s_t)$ is estimated by a lightweight proxy $\widehat{U_t}$ computed from the $B_t$ candidates at step $t$ (see Equation 6).

**Uncertainty-Aware Pruning (UAP)** UAP aims to prune redundant candidate actions based on the step-wise uncertainty $U_t$. Specifically, when $U_t$ is below a threshold $\lambda$, only the top-$\hat{k}$ candidates with the highest value $V(s_t, a_t^{(i)})$ are kept in the search tree as the effective children so as to avoid exploring necessary reasoning trajectories. Otherwise we keep all candidates to ensure the diversity of reasoning trajectories and the final answers:

$$\widehat{\mathcal{C}}_t = \begin{cases} \arg\text{top-}\hat{k}_{a_t^{(i)} \in \mathcal{C}_t} V\left(s_t, a_t^{(i)}\right), & U_t < \lambda, \\ \mathcal{C}_t, & \text{otherwise,} \end{cases}$$

where $\widehat{k} < B$ is the number of kept candidate actions and $\lambda$ is the uncertainty threshold.

**Uncertainty-Aware Budgeting (UAB)**  For each retained action $a_t^{(i)} \in \widehat{\mathcal{C}}_t$, UAB assigns a next-step expansion budget that depends only on the parent's uncertainty $U_t$:

$$B_{t+1} := \begin{cases} \widehat{B}, & U_t < \lambda, \\ B, & \text{otherwise,} \end{cases} \tag{5}$$

where $0 < \widehat{B} < B$ and $B$ is the user-specific expansion budget of the base algorithm. Thus, low-uncertainty parents use the smaller budget $\widehat{B}$ to avoid redundant exploration, whereas high-uncertainty parents keep the full budget $B$.

**Approximation of reasoning uncertainty**  Since directly computing $U(A_t \mid S_t = s_t)$ is infeasible, we approximate it by sampling multiple candidate actions, following prior work (Lin et al., 2023). Specifically, we estimate the uncertainty using the dispersion of embeddings of the generated candidates $\{a_t^{(i)}\}_{i=1}^{B_t}$. Concretely, we define

$$\widehat{U}(A_t \mid S_t = s_t) = \frac{2}{B_t(B_t - 1)} \sum_{1 \leq i < j \leq B_t} \big(1 - \cos(e_i, e_j)\big), \tag{6}$$

where $e_i$ is the embedding of action $a_t^{(i)}$, and $\cos(e_i, e_j)$ denotes the cosine similarity. Equation 6 reflects the average pairwise dissimilarity among candidates, which serves as a proxy for uncertainty. In our main experiments (Section 5), we obtain the embeddings by extracting the feature vector of the last token of each generated action, which provides a compact representation of its semantics.

## 5 EXPERIMENTS

### 5.1 EXPERIMENT SETUP

**Models and Datasets**  In our experiments, we evaluate our method on mathematical reasoning. A, we choose *Llama3.1-8B-Instruct* (Grattafiori et al., 2024) and *Qwen3-32B* (Yang et al., 2025), selected for its strong instruction-following and reasoning capabilities. Inference is conducted via *SGLang* (Zheng et al., 2024), with temperature set to 0.8 for *Llama3.1-8B-Instruct* and 0.7 for *Qwen3-32B* and other parameters at default. Experiments were run on two NVIDIA NVL H100 94GB GPUs. For datasets, we use three representative benchmarks, including **GSM8K** (Cobbe et al., 2021), **MATH500** (Lightman et al., 2023), (Hendrycks et al.), and **AIME24**. Due to the high computational cost of tree-based reasoning algorithms, we conduct experiments on the first 100 examples from GSM8K and MATH500, following the experimental protocol in (Wang et al., 2025b).

**Methods**  We evaluate UAA on Beam Search and MCTS (denoted as +UAA), with implementation details in Appendix F. For comparison, we also include **LiteSearch** (Wang et al., 2024a), which improves efficiency via step-aware selection and reduced budgets at deeper levels.

**Value functions**  In this work, we evaluate two types of value functions to guide search. (1) *LLM confidence*, (dubbed Conf.), computed as in Equation 2 and rescaled to the $[0, 1]$ range via exponential transformation to align with LiteSearch; (2) *Process reward model*, (dubbed PRM), fine-tuned from Llama3.1-8B-Instruct using RLHFlow and the Deepseek-PRM-Data (Dong et al., 2024).

**Metrics**  For evaluation, we report three metrics: (i) *Accuracy*, computed by exact match after majority voting across reasoning paths.[1] (ii) *Average tokens*, serving as a proxy for the computational cost; and (iii) *Time*, the wall-clock time required to generate the entire reasoning tree. We repeat each experiment three times and report the mean value of each metric.

### 5.2 MAIN RESULTS

**UAA improves the reasoning efficiency of Beam search.**  To evaluate the effectiveness of UAA, we apply it to beam search under two types of value functions: *Conf.* and *PRM*. We set the number of beams to 16, the tree width to 4, and the maximum tree depth to 40. We fix the uncertainty threshold

---

[1]We use Qwen's official answer verification tool for consistency: https://github.com/QwenLM/Qwen2.5-Math/tree/main. Evaluation results can vary with the checker; for reproducibility, we recommend using the same implementation.

Table 1: Main results of Beam Search and MCTS on *Llama3.1-8B-Instruct* across GSM8K, MATH-500, and AIME24. Metrics are reported as mean values.

| Verifier | Method | GSM8K | | | MATH-500 | | | AIME24 | | |
|---|---|---|---|---|---|---|---|---|---|---|
| | | Acc. (%) ↑ | Tokens $(k)$ ↓ | Time $(s)$ ↓ | Acc. (%) ↑ | Tokens $(k)$ ↓ | Time $(s)$ ↓ | Acc. (%) ↑ | Tokens $(k)$ ↓ | Time $(s)$ ↓ |
| Conf. | LiteSearch | 80.14 | 0.71 | 14.97 | 54.67 | 2.06 | 38.57 | 5.56 | 5.25 | 92.08 |
| | Beam Search | 89.17 | 3.12 | 53.32 | 57.17 | 11.90 | 234.88 | 6.67 | 35.97 | 734.44 |
| | +UAA | 88.00 | **2.58** | 34.23 | 56.40 | **8.27** | 126.21 | 6.67 | **20.96** | 341.69 |
| | MCTS | 90.17 | 8.22 | 114.75 | 61.50 | 21.65 | 290.40 | 7.78 | 53.09 | 1261.83 |
| | +UAA | 90.50 | **5.43** | 74.58 | 61.50 | **11.02** | 175.60 | 6.11 | **19.02** | 414.60 |
| PRM | LiteSearch | 85.67 | 0.73 | 731.14 | 58.00 | 2.27 | 42.10 | 5.56 | 6.54 | 100.55 |
| | Beam Search | 91.33 | 2.84 | 49.57 | 59.33 | 8.13 | 150.59 | 8.89 | 20.82 | 411.04 |
| | +UAA | 90.00 | **2.21** | 29.45 | 58.00 | **7.34** | 112.73 | 6.67 | **18.13** | 299.58 |
| | MCTS | 92.00 | 8.94 | 123.73 | 63.00 | 17.63 | 255.95 | 8.89 | 33.46 | 581.19 |
| | +UAA | 91.33 | **6.20** | 86.57 | 65.00 | **10.93** | 172.87 | 8.89 | **16.41** | 369.50 |

at $\lambda = 0.5$ and set $\hat{k} = 2$ and $\widehat{B} = 3$. As shown in Table 1, UAA consistently improves the inference efficiency of beam search across all datasets and verifier settings, while maintaining or even improving accuracy. For example, on GSM8K with Conf., the UAA achieves 88.00% accuracy using only 2.58k tokens and 34.23 seconds, outperforming both the base beam search (89.17%, 3.12k tokens and 53.32s ). Similar trends hold on MATH500 and AIME24, confirming that UAA effectively reduces redundant computation while retaining a comparable performance. Moreover, the results of *Qwen3-32B* is presented in Appendix G. It is also worth noting that while LiteSearch achieves the lowest token usage, it often results in lower accuracy, demonstrating a trade-off between efficiency and performance that UAA successfully balances. More results on a realistic planning dataset Blocksworld (Valmeekam et al., 2023) and a common sense logical reasoning dataset StrategyQA (Geva et al., 2021) is provided in Appendix H.1.

**UAA improves the reasoning efficiency of MCTS.** We evaluate UAA on MCTS using the open-source library *OpenR* (Wang et al., 2024c). Specifically, we set the tree width to $B = 4$, the maximum depth to 40, and the number of iterations to 100. For UAA, We fix the uncertainty threshold at $\lambda = 0.5$ and set $\hat{k} = 2$ and $\widehat{B} = 2$. Table 1 presents the experimental results of the standard MCTS and MCTS with UAA. Across all datasets and verification strategies, UAA consistently reduces token usage and inference time while maintaining comparable accuracy. For example, on the GSM8K dataset with *Conf.* verifier, UAA drops tokens from 8.22k to 5.43k and time from 114.75s to 74.58s, but maintains a comparable generation accuracy. These results demonstrate that UAA effectively reduces the computational cost in MCTS without sacrificing prediction quality.

**Ablation study** We conduct ablation studies for two factors of UAA on GSM8K: the uncertainty threshold $\lambda$ and the two components: uncertainty-aware pruning (UAP) and uncertainty-aware budgeting (UAB). Figure 3 shows that across Beam Search and MCTS with both verifiers (Conf. and PRM), increasing $\lambda$ monotonically reduces token usage while keeping accuracy essentially unchanged. The blue curves drop as $\lambda$ grows, whereas the red accuracy curves remain flat and close to the base accuracy. This shows that stronger pruning guided by higher uncertainty thresholds cuts redundant expansions without harming solution quality.

Table 2: Ablation study of UAA on GSM8K.

| Verifier | UAP | UAB | Acc. (%) ↑ | Tokens $(k)$ ↓ | Time (s) |
|---|---|---|---|---|---|
| Conf. | ✗ | ✗ | 89.17 | 3.12 | 53.32 |
| | ✓ | ✗ | 87.50 | 2.69 | 37.05 |
| | ✗ | ✓ | 86.50 | 2.63 | 33.69 |
| | ✓ | ✓ | 88.00 | **2.58** | 34.23 |
| PRM | ✗ | ✗ | 91.33 | 2.84 | 49.57 |
| | ✓ | ✗ | 90.00 | 2.36 | 34.33 |
| | ✗ | ✓ | 89.50 | 2.22 | 29.88 |
| | ✓ | ✓ | 90.00 | **2.21** | 29.45 |

Table 2 isolates the contributions of UAP and UAB. With the confidence verifier, UAB alone uses fewer tokens than UAP alone (2.63k vs. 2.69k), whereas UAP alone attains higher accuracy (87.50% vs. 86.50%). Combining the two yields the best trade-off: 88.00% accuracy with the fewest tokens (2.58k) and near-best time (34.23s), which reduces tokens relative to UAP only and UAB only. A similar trend can be found for the PRM verifier. Overall, the threshold $\lambda$ governs a smooth efficiency–accuracy trade-off, and UAP and UAB are complementary: UAB delivers most of the cost savings, UAP focuses the limited budget on high-value trajectories, and together they achieve the best efficiency while maintaining accuracy. Moreover, we provide a depth analysis of our framework in Appendix I.

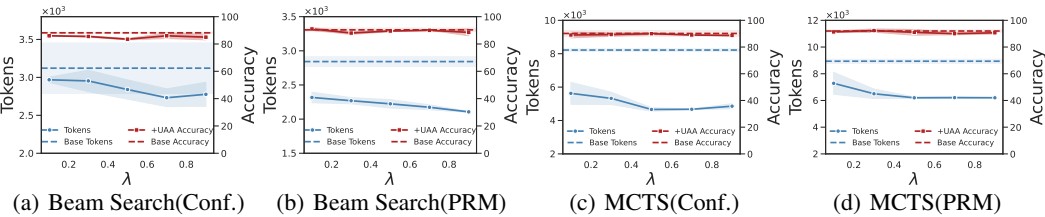

(a) Beam Search(Conf.)    (b) Beam Search(PRM)    (c) MCTS(Conf.)    (d) MCTS(PRM)

Figure 3: Ablation of the uncertainty threshold $\lambda$ on GSM8K. (a–b) report Beam Search, and (c–d) report MCTS, using either model confidence (Conf.) or process reward (PRM) as the value function. Blue curves show token usage (left axis) and red curves show accuracy (right axis).

Table 3: Experimental results of Beam Search on *Llama3.1-8B-Instruct* with different uncertainty estimators. Entries under the "Uncertainty Type" column include "Conf." and "PRM," which denote uncertainty scores based on the variance of confidence and process reward values, respectively, while "Embedding" is derived from the dispersion of action embeddings.

| Verifier | Uncertainty Type | GSM8K | | | MATH-500 | | | AIME24 | | |
|---|---|---|---|---|---|---|---|---|---|---|
| | | Acc. (%) ↑ | Tokens (k) ↓ | Time (s) ↓ | Acc. (%) ↑ | Tokens (k) ↓ | Time (s) ↓ | Acc. (%) ↑ | Tokens (k) ↓ | Time (s) ↓ |
| Conf. | Conf. | 85.50 | 2.86 | 38.02 | 56.67 | **8.01** | 123.43 | 5.56 | **20.24** | 332.61 |
| | PRM | 87.00 | 2.63 | 36.38 | 54.00 | 8.70 | 135.46 | 6.67 | 21.61 | 360.51 |
| | Embedding | **88.00** | **2.58** | **34.23** | **56.40** | 8.27 | 126.21 | 6.67 | 20.96 | 341.69 |
| PRM | Conf. | 86.00 | 2.73 | 37.67 | 58.33 | 8.61 | 142.79 | 6.67 | 18.81 | 318.28 |
| | PRM | 87.50 | 2.40 | 33.97 | 57.00 | 7.99 | 132.81 | 6.67 | 20.26 | 333.34 |
| | Embedding | **90.00** | **2.21** | **29.45** | 58.00 | **7.34** | **112.73** | 6.67 | **18.13** | **299.58** |

Table 4: Experimental results of DVTS, REBASE, and their combinations with UAA on GSM8K, MATH500, and AIME24 benchmarks with *Llama3.1-8B-Instruct*.

| Method | | GSM8K | | | MATH500 | | | AIME24 | | |
|---|---|---|---|---|---|---|---|---|---|---|
| | | Acc. (%) ↑ | Tokens (k) ↓ | Time (s) ↓ | Acc. (%) ↑ | Tokens (k) ↓ | Time (s) ↓ | Acc. (%) ↑ | Tokens (k) ↓ | Time (s) ↓ |
| DVTS | Base | 87.3 | 3.4 | 75.5 | 60.7 | 12.1 | 408.1 | 6.7 | 31.8 | 973.6 |
| | +UAA | 88.3 | **1.9** | **48.0** | 59.0 | **4.9** | **99.4** | 7.8 | **11.3** | **243.1** |
| REBASE | Base | 88.7 | 5.1 | 191.8 | 58.3 | 12.4 | 488.1 | 7.8 | 27.2 | 995.6 |
| | +UAA | 88.7 | **4.7** | **153.8** | 61.0 | **11.7** | **436.4** | 8.9 | **26.2** | **790.6** |

> **Takeaways:** 1. UAA reduces computational cost for Beam Search/MCTS without hurting accuracy. 2. The threshold $\lambda$ induces a smooth efficiency–accuracy trade-off. UAP and UAB are complementary: UAB provides most cost savings, UAP focuses the limited budget on high-value trajectories, and together they deliver the best efficiency without sacrificing accuracy.

# 6 DISCUSSION

**Empirical validation of UAA under black-box settings** In Section 4, we used the semantic uncertainty of action embeddings as the uncertainty estimator. However, in real-world black-box scenarios, embedding vectors of the generated tokens are often unavailable, especially for closed-source LLMs. To address this, we evaluate two proxies: (i) the variance of confidence scores and (ii) the variance of PRM scores. We evaluate the effectiveness of these two proxies on Beam Search with *Llama3.1-8B-Instruct* under the same hyperparameters in Section 5. As shown in Table 3, the embedding-based estimator is generally the strongest, especially when PRM is used as the verifier, e.g., on GSM8K it attains 90.0% accuracy with 2.21k tokens. In some cases, the proxies match or even slightly outperform embeddings. For example, on MATH-500 with the confidence verifier, the confidence-variance proxy achieves 56.67% accuracy with 8.01k tokens, compared to 56.40% and 8.27k for the embedding estimator. In summary, combining the baseline results in Table 1 with the ablations in Table 3, we find that embedding dispersion is generally the most reliable estimator of semantic uncertainty, while variance over confidence or PRM scores remains a practical fallback in black-box settings where embeddings are inaccessible. The results of GPT-4o can be found in Appendix H.2.

**UAA can improve advanced tree search algorithms** We compare UAA with two advanced tree-search methods, DVTS (Beeching et al.) and REBASE (Wu et al., 2024). DVTS improves diversity by splitting the beam into verifier-guided subtrees, while REBASE balances expansion using reward-based budget allocation. We integrate UAA into both methods (implementation details in Appendix F) and evaluate them under *Llama3.1-8B-Instruct* with confidence scores as value guidance. For UAA, we fix the uncertainty threshold at $\lambda = 0.5$, $\hat{k} = 16$ and $\widehat{B} = 1$. As shown in Table 4, UAA consistently reduces token usage and wall-clock time without harming accuracy. For instance, on GSM8K, DVTS+UAA reaches 88.3% accuracy with 1.9k tokens and 48.0s, compared to 87.3%, 3.4k tokens and 75.5s for base DVTS. Similar gains are observed on GSM8K and MATH500, demonstrating the complementarity of UAA with existing methods.

## 7 RELATED WORK

**Efficient tree-search reasoning of LLMs** Tree-search–based reasoning methods (Yao et al., 2023; Feng et al.; Wang et al., 2024b; Chen et al., 2024) have substantially advanced the reasoning capabilities of LLMs. To further improve efficiency, *LiteSearch* (Wang et al., 2024a) proposes a step-aware selection rule and dynamically expands children under a budgeted computation scheme. *Uncertainty-guided Likelihood-Tree Search* (Grosse et al., 2024) adopts a token-level strategy that leverages probabilistic uncertainty to guide non-myopic search, yielding better sample efficiency and likelihood optimization than beam search; however, it is inherently tailored to likelihood-based objectives and does not directly accommodate reward-driven generation. *FETCH* (Wang et al., 2025a) merges states after each expansion via agglomerative clustering with a post-trained embedding model, introducing extra overhead for embedding extraction and clustering. Uncertainty-Aware Modeling (Yu et al., 2025) integrates uncertainty into value models to guide candidate selection during search. *EquivPruner* (Liu et al., 2025) trains an equivalence detector to detect and remove semantically equivalent actions. Meanwhile, *SpecSearch* (Wang et al., 2025b) reduces end-to-end latency via speculative mechanisms but relies on an auxiliary draft pathway, increasing resource usage. Different from these approaches, we propose a *training-free*, *plug-and-play* framework that lowers computational cost in terms of both tokens and latency, requires no auxiliary models or fine-tuning, and is applicable to both *black-box* and *white-box* LLM settings.

**Uncertainty estimation in LLMs** As LLMs continue to evolve, understanding and quantifying the uncertainty in their predictions is critical for enhancing application credibility and mitigating hallucinations. A common strategy is to quantify uncertainty from multiple generated responses, offering both performance gains and a means to assess confidence. Ensemble-based methods, such as variance, consistency, and similarity-based techniques (Fadeeva et al., 2023; Wang et al., 2022; Lin et al., 2023), quantify uncertainty by analyzing disagreement across sampled outputs. Recent works also apply conformal prediction (Huang et al., 2023a; Cherian et al., 2024; Yadkori et al., 2024) to provide set-valued predictions with formal coverage guarantees. Other studies, like Uncertainty-of-Thoughts (Hu et al., 2024), leverage uncertainty to guide information-seeking behavior during reasoning. Inspired by predictive uncertainty quantification, We propose a novel uncertainty-aware tree search framework that allocates computation adaptively based on model confidence.

## 8 CONCLUSION

We proposed Uncertainty-Aware Allocation (UAA), a simple, training-free framework that improves the efficiency of tree-search reasoning for LLMs. By pruning redundant actions and shrinking budgets at low-uncertainty steps, UAA significantly reduces token usage and inference time while maintaining accuracy across Beam Search, MCTS, and multiple benchmarks. The key insight is that sampling low-uncertainty steps with the same budget as high-uncertainty ones is redundant. We hope the insights and analyses presented here will inspire further advances in reasoning and training strategies for LLMs. A promising direction for future work is to extend this idea to LLM-powered agent planning, where uncertainty-guided budgeting may further improve planning efficiency.

**Limitations** The current design of UAA adopts fixed thresholds and budgets for compute allocation, which restricts its adaptability across reasoning depths. In addition, its generalization to open-domain tasks remains an open question.

REPRODUCIBILITY STATEMENT

We have made every effort to ensure that the results presented in this paper are reproducible. The key code and datasets have been made publicly available in an anonymous repository to facilitate replication and verification. The experimental setup, including model configurations and hardware details, is described in detail in the paper. We have also provided a full description to assist others in reproducing our experiments.

Additionally, the datasets used in this work are publicly available, ensuring consistent and reproducible evaluation results. We believe these measures will enable other researchers to reproduce our work and further advance the field.

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

# A    LLM USAGE

This manuscript was developed with the assistance of large language models, which helped enhance the clarity, coherence, and technical precision of the writing. The authors' original research contributions, intellectual property, and key arguments remain entirely their own. All content has undergone thorough review and validation by the authors to ensure accuracy and scientific integrity.

# B    DETAILS OF PILOT STUDY

In this section, we provide the experimental details of computing the pairwise similarity among sampled continuations. Specifically, for each state $s_t$, we embed the $m$ sampled continuations $\{p^c_{s_t,i}\}^m_{i=1}$ using a pretrained text encoder to obtain vectors $\{e_i\}^m_{i=1}$. To quantify the diversity of final answers, we simply count the number of distinct solutions among the produced answers.

## B.1    COMPUTATION OF TEXT SIMILARITY

Let the candidate set of paths at decoding step $t$ be denoted as $\{p^c_{s_t,1}, p^c_{s_t,2}, \ldots, p^c_{s_t,m}\}$, where each path is represented by an embedding vector $e(p^c_{s_t,i}) \in \mathbb{R}^d$. We organize these embeddings into a matrix:

$$E = \begin{bmatrix} e(p^c_{s_t,1}) \\ e(p^c_{s_t,2}) \\ \vdots \\ e(p^c_{s_t,m}) \end{bmatrix} \in \mathbb{R}^{m \times d}$$

To quantify the overall similarity among these candidate embeddings, we compute the average pairwise cosine similarity as follows:

1. Normalize each embedding vector to unit length to obtain $\tilde{E} \in \mathbb{R}^{m \times d}$:

$$\tilde{e}_i = \frac{e(p^c_{s_t,i})}{\|e(p^c_{s_t,i})\|_2 + \varepsilon}$$

   where $\varepsilon$ is a small constant for numerical stability.

2. Compute the cosine similarity matrix:

$$\mathcal{E} = \tilde{E}\tilde{E}^\top \in \mathbb{R}^{m \times m}$$

   where $\mathcal{E}_{ij}$ denotes the cosine similarity between embeddings $i$ and $j$.

3. Exclude the diagonal (self-similarity) entries and compute the average of all off-diagonal elements:

$$\text{Sim}(p^c_{s_t,1}, p^c_{s_t,2}, \ldots, p^c_{s_t,m}) = \frac{1}{m(m-1)} \sum_{\substack{i,j=1 \\ i \neq j}}^{m} \mathcal{E}_{ij}$$

This metric reflects the internal consistency among the candidate paths. A higher value indicates that the candidates are more similar to each other in the embedding space.

## B.2    DETAILS OF MOTIVATION EXPERIMENT

We generate reasoning trajectories using a reward-guided beam search strategy (Yao et al., 2023). Experiments are conducted on a subset of 100 samples from the GSM8K benchmark (Cobbe et al., 2021), using the LLaMA3.1-8B-Instruct model (Grattafiori et al., 2024). At each decoding step, 20 candidate continuations are sampled from every active node, and the top 20 are selected based on reward scores provided by the Skywork-o1-Open-PRM-Qwen-2.5-1.5B model[2]. The beam width is fixed at 20 throughout the search. This setup yields a wide and diverse reasoning tree, enabling fine-grained analysis of reasoning dynamics. To compute semantic uncertainty, we embed each sampled continuation using the gte-Qwen2-7B-instruct encoder (Li et al., 2023) and estimate the embedding dispersion by one minus the semantic similarity.

---

[2] https://huggingface.co/Skywork/Skywork-o1-Open-PRM-Qwen-2.5-1.5B

### B.3 EMPIRICAL ANALYSIS BETWEEN CURRENT REASONING STEPS AND CONTINUATIONS

Following the previous pilot study, we further investigate the relationship between a reasoning step and the semantic redundancy of its downstream continuations. Specifically, we examine whether the semantic similarity among sibling nodes at a given step correlates with the similarity of their respective continuations.

As shown in Figure 4, the distribution of Spearman correlations across multiple steps reveals a consistently positive correlation between the similarity of sibling nodes and the similarity of their continuations. This empirical finding confirms that **semantic redundancy at the current step is predictive of redundancy in downstream reasoning paths**.

These results provide further motivation for our proposed approach: **by identifying low-uncertainty steps in the search tree, we can safely reduce the search budget allocated to their continuations without sacrificing solution diversity.**

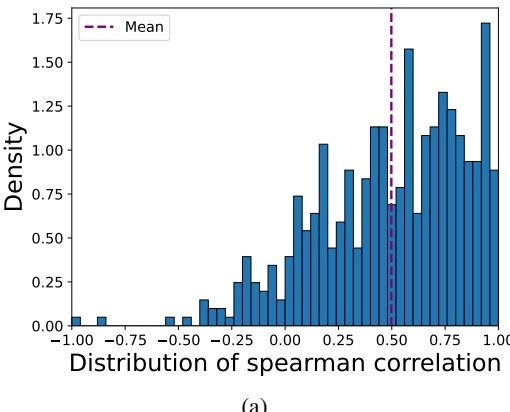

(a)

Figure 4: Spearman correlation between the dispersion of continuations and the dispersion of sibling nodes.

## C MORE EMPIRICAL ANALYSIS

In this part, we further investigate the generality of the proposed assumption across MCTS algorithms, large LLMs, and open-domain tasks, i.e, StrategyQA (Geva et al., 2021) . Experimental settings not explicitly specified here follow those in Appendix B.2.

**Empirical study on an open-domain task** In this experiment, we evaluate Beam Search on the open-domain reasoning task, i.e., StrategyQA, using *LLaMA3.1-8B-Instruct* and *Qwen3-32B* under the same decoding configuration. Since PRM are not applicable to StrategyQA, we adopt LLM confidence as the guidance signal during reasoning. The empirical results are shown in Figures 5 and 6. As illustrated, the reasoning trajectories across beams exhibit extremely high textual similarity, with most similarities exceeding 0.95, indicating that beam exploration collapses to nearly identical reasoning patterns. Furthermore, the Spearman correlation between trajectory similarity and the diversity of final answers is weak and often negative, suggesting that even highly overlapping reasoning paths can lead to inconsistent predictions. These observations demonstrate that, on open-domain tasks like StrategyQA, expanding the search space via Beam Search offers limited semantic diversification, and the resulting trajectory consistency does not translate into stable answer outputs.

**Empirical study on MCTS** We conduct a pilot study of MCTS on the GSM8K dataset using the *LLaMA3.1-8B-Instruct* and *Qwen3-32B* models. Specifically, we set the tree width to $B = 4$, the maximum depth to 40, and the number of iterations to 100. The results are presented in Figure 7 and 8. As shown, we observe substantial redundancy in the generated reasoning trajectories. In particular, reasoning steps that are semantically similar contribute little to expanding the coverage of diverse answer candidates.

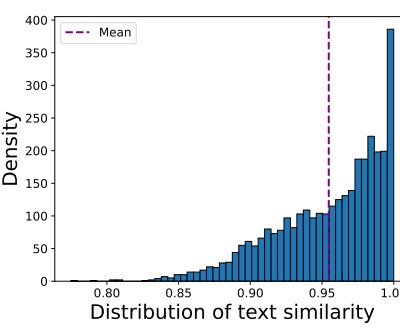 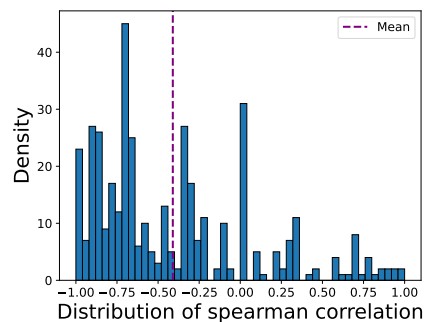

(a) Similarity of reasoning trajectories

(b) Spearman correlation between similarities of reasoning trajectories and diversities of final answers

Figure 5: Empirical study of Beam Search on StrategyQA Using *LLaMA3.1-8B-Instruct*.

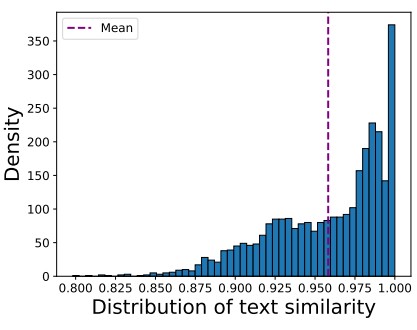 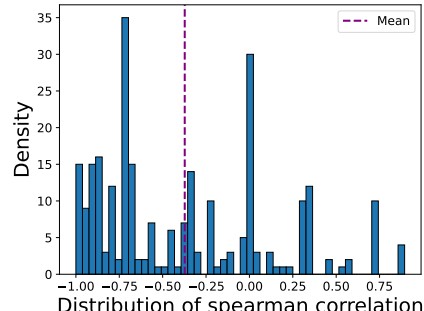

(a) Similarity of reasoning trajectories

(b) Spearman correlation between similarities of reasoning trajectories and diversities of final answers

Figure 6: Empirical study of Beam Search on StrategyQA Using *Qwen3-32B*.

**Empirical study for Larger LLM** We conduct a pilot investigation of both MCTS and Beam Search on GSM8K using the *Qwen3-32B* model. The corresponding results are illustrated in Figure 8 and Figure 9. Across both search strategies, we observe a pronounced redundancy within the generated reasoning trajectories: the text-similarity distributions are heavily skewed toward near-identical reasoning paths, and the Spearman correlations indicate that such similarity contributes minimally to diversifying final predictions. Similar results can also be found in the empirical study of StrategyQA on Qwen3-32B, shown in Figure 6. These findings suggest that even with larger LLMs, the search process tends to collapse into highly similar reasoning branches, limiting its ability to explore meaningfully distinct solution candidates.

## D EMPIRICAL EXPERIMENT ABOUT COROLLARY 1

To complement the theoretical result in Corollary 1, we conduct an auxiliary analysis on Beam Search to test whether the semantic dispersion of candidate actions correlates with downstream answer accuracy. The experiment setting is the same as shown in Appendix B.2. For each child node $c$, we compute (i) its semantic dispersion, defined as the mean pairwise distance among $c$'s own child nodes (i.e., the candidate actions generated when expanding $c$), and (ii) the average accuracy of the resulting final answers. Within each parent node, we then compute the Spearman rank correlation between dispersion and accuracy across its children.

The results, shown in Figure 10, show that across all datasets and reasoning steps, the distribution of correlations is predominantly positive, with the median value significantly above zero. This indicates

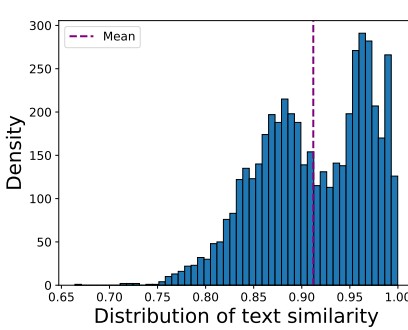 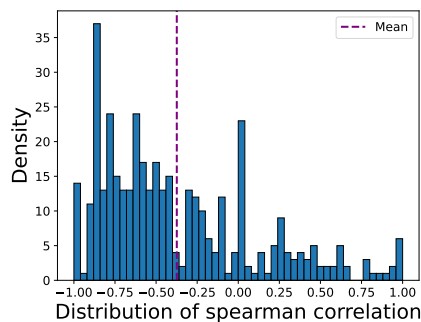

(a) Similarity of reasoning trajectories

(b) Spearman correlation between similarities of reasoning trajectories and diversities of final answers

Figure 7: Empirical study of MCTS on GSM8K Using *LLaMA3.1-8B-Instruct*.

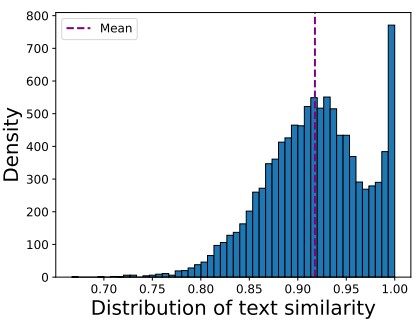 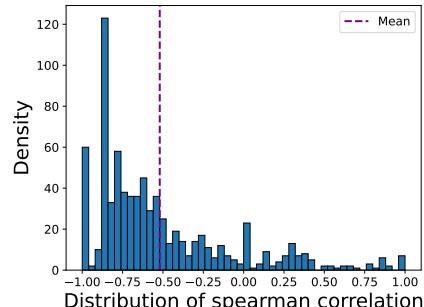

(a) Similarity of reasoning trajectories

(b) Spearman correlation between similarities of reasoning trajectories and diversities of final answers

Figure 8: Empirical study of MCTS on GSM8K Using *Qwen3-32B*.

that children exhibiting higher semantic dispersion tend to achieve higher accuracy, suggesting that additional exploration is more beneficial in nodes where the sampled actions are more diverse. **Conversely, sibling nodes with highly redundant actions show limited marginal gains on the final solution quality under the same budget.** These empirical findings are consistent with the theoretical results: **greater uncertainty requires more samples to provide more opportunity for accuracy improvements**.

# E PROOFS OF THEORETICAL ANALYSIS

**Proposition.** *Let $Y$ be a target variable, and let $A_t \sim \pi_\theta(\cdot \mid s_t)$ denote a random action sampled from the policy at state $s_t$. Define the conditional entropy as $H(Y \mid A_t, s_t) := \mathbb{E}_{a_t \sim \pi_\theta}[h(a_t)]$, where $h(a_t) := -\sum_y p(y \mid a_t, s_t) \log p(y \mid a_t, s_t)$. Let $\hat{H}_n := \frac{1}{n} \sum_{i=1}^n h(a_t^{(i)})$ be the average of i.i.d. samples $\{a_t^{(i)}\}_{i=1}^n \sim \pi_\theta(\cdot \mid S_t = s_t)$. If $h(\cdot)$ is $C$-Lipschitz, then for any $\varepsilon > 0$,*

$$\mathbb{P}\big(\big|\hat{H}_n - H(Y \mid A_t, s_t)\big| \geq \varepsilon\big) \leq \frac{C^2 \cdot \mathrm{Var}(A_t \mid s_t)}{n\varepsilon^2}.$$

*where $\mathrm{Var}(A_t \mid s_t)$ is the variance (i.e., uncertainty ) of $A_t$ conditional on $s_t$.*

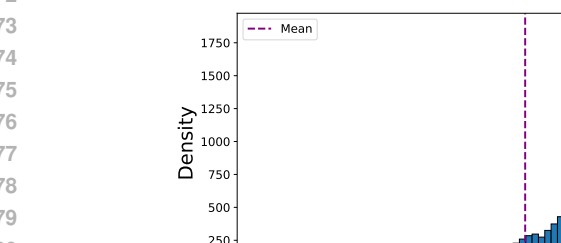 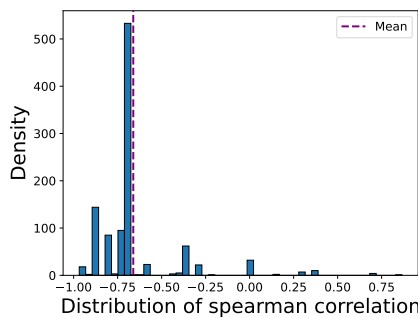

(a) Similarity of reasoning trajectories

(b) Spearman correlation between similarities of reasoning trajectories and diversities of final answers

Figure 9: Empirical study of Beam Search on GSM8K Using *Qwen3-32B*.

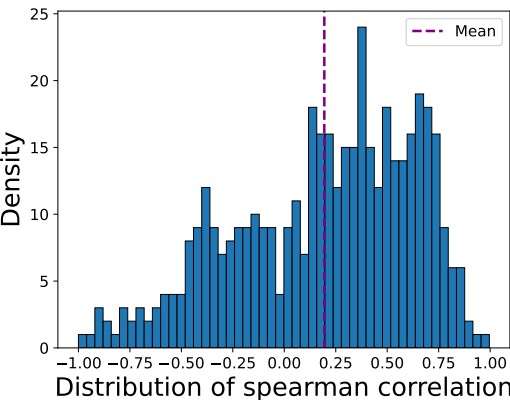

Figure 10: Correlation between semantic uncertainty of sibling nodes and answer accuracy.

*Proof.* Let $h(a_t) := H(Y \mid A_t = a_t, s_t)$, and define the true expectation and estimator as:

$$H(Y \mid A_t, s_t) = \mathbb{E}_{a_t \sim \pi_\theta}[h(a_t)], \quad \hat{H}_n = \frac{1}{n} \sum_{i=1}^{n} h(a_t^{(i)}),$$

where $a_t^{(i)} \sim \pi_\theta(\cdot \mid s_t)$ are i.i.d. samples.

Since $\hat{H}_n$ is an empirical mean of i.i.d. random variables $h(a_t^{(i)})$, we have:

$$\mathbb{E}[\hat{H}_n] = \mathbb{E}[h(A_t)] = H(Y \mid A_t, s_t), \quad \text{so } \hat{H}_n \text{ is unbiased.}$$

Let us compute the variance of $\hat{H}_n$:

$$\mathrm{Var}(\hat{H}_n) = \mathrm{Var}\left(\frac{1}{n} \sum_{i=1}^{n} h(a_t^{(i)})\right) = \frac{1}{n} \mathrm{Var}(h(A_t)).$$

Now apply the Lipschitz condition. Since $h$ is $C$-Lipschitz in $a_t$, by the standard result on Lipschitz functions of random variables:

$$\mathrm{Var}(h(A_t)) \leq C^2 \cdot \mathrm{Var}(A_t \mid s_t).$$

Therefore,

$$\mathrm{Var}(\hat{H}_n) \leq \frac{C^2}{n} \cdot \mathrm{Var}(A_t \mid s_t).$$

Finally, applying Chebyshev's inequality for any $\varepsilon > 0$,

$$\mathbb{P}\left(\left|\hat{H}_n - \mathbb{E}[\hat{H}_n]\right| \geq \varepsilon\right) \leq \frac{\mathrm{Var}(\hat{H}_n)}{\varepsilon^2} \leq \frac{C^2 \cdot \mathrm{Var}(A_t \mid s_t)}{n\varepsilon^2}.$$

Since $\mathbb{E}[\hat{H}_n] = H(Y \mid A_t, s_t)$, we conclude:

$$\mathbb{P}\left(\left|\hat{H}_n - H(Y \mid A_t, s_t)\right| \geq \varepsilon\right) \leq \frac{C^2 \cdot \mathrm{Var}(A_t \mid s_t)}{n\varepsilon^2}.$$

$\square$

## F  EXPERIMENTAL DETAILS

### F.1  DETAILS OF TREE SEARCH ALGORITHMS

#### F.1.1  BEAM SEARCH

Beam search optimizes the value of reasoning trajectories by searching over its per-step predictions. Our implementation is similar to value-guided Breadth First Search (Feng et al.; Yao et al., 2023; Snell et al., 2024; Beeching et al.). Concretely, we consider a fixed number of beams $N$ and a beam width $M$. We then run the following steps:

1. Sample $N$ initial predictions for the first step in the solution.

2. Score the generated steps according to the PRM's predicted step-wise reward-to-go estimate (which also corresponds to the total reward from the prefix since the reward is sparse in this setting).

3. Filter for only the top $\frac{N}{M}$ highest scoring steps.

4. From each candidate, sample $M$ proposals from the next step, resulting in a total of $\frac{N}{M} \times M$ candidate prefixes again. Then repeat steps 2–4.

**Incorporating UAA into Beam Search.**  We integrate UAA into the above process by modifying Steps 3 and 4. In Step 3, *Uncertainty-Aware Pruning* (UAP) removes redundant candidates based on their uncertainty estimates. In Step 4, *Uncertainty-Aware Budgeting* (UAB) allocates the expansion budget across the retained candidates. Moreover, to enhance the stability of UAA across different datasets and scales, we apply min-max normalization to the estimated uncertainty values during each expansion step. Specifically, following the notation in Section 2, let $\{s_t^{(l)}\}_{l=1}^L$ be the set of candidate states at step $t$. We compute their corresponding uncertainty estimates

$$\widehat{U}'(A_t \mid S_t = s_t^{(l)}) = \frac{\widehat{U}(A_t \mid S_t = s_t^{(l)}) - \min_l \widehat{U}(A_t \mid S_t = s_t^{(l)})}{\max_i \widehat{U}(A_t \mid S_t = s_t^{(l)}) - \min_l \widehat{U}(A_t \mid S_t = s_t^{(l)})}. \tag{7}$$

#### F.1.2  MONTE CARLO TREE SEARCH (MCTS)

MCTS (Feng et al.; Wang et al., 2024b) is a best-first search algorithm that incrementally builds a search tree by balancing exploration and exploitation through Monte Carlo simulations. It consists of four main stages:

1. **Selection:** Starting from the root node, the algorithm selects child nodes based on the Upper Confidence Bound (UCB) criterion:

$$s_{t+1} = \arg\max_{s_j \in \mathrm{Children}(s_t)} \left[ V(s_j) + c \cdot \sqrt{\frac{\log N(s_t)}{1 + N(s_j)}} \right],$$

where $V(s_j)$ is the value of node $s_j$, $N(s_t)$ is the visit count of the parent node, and $c$ is a constant balancing exploration and exploitation. This selection continues until a leaf node is reached.

2. **Expansion:** If the selected leaf node is non-terminal (i.e., no final answer has been generated), it is expanded by decoding one step using the policy model to generate $k$ child nodes.

3. **Simulation:** From each expanded node $s_t$, the policy model performs $B$ rollouts to generate complete reasoning trajectories, which are then scored by a reward model. The initial value of $s_c$ is given by a value function.

4. **Backpropagation:** Once the child nodes are evaluated, their values are propagated up the tree. The parent node $s_t$ updates its value and visit count as follows:

$$V(s_t) \leftarrow \frac{N(s_t) \cdot V(s_t) + \sum_{c=1}^{k} V(s_c)}{N(s_t) + k}, \quad N(s_t) \leftarrow N(s_t) + 1.$$

**Incorporating UAA into MCTS.** We integrate UAA into MCTS to improve efficiency. In the Selection stage, *Uncertainty-Aware Pruning* (UAP) is applied to filter out candidate actions with low uncertainty and low values, thereby focusing the search on more promising nodes. During the Expansion stage, *Uncertainty-Aware Budgeting* (UAB) adaptively allocates the expansion budget, deciding how many child nodes to expand for each selected node according to their uncertainty levels.

### F.1.3 DIVERSE VERIFIER TREE SEARCH (DVTS)

DVTS (Beeching et al.) is an extension of beam search designed to maximize diversity at large test-time compute budgets. It follows a similar procedure to beam search, with the following modifications:

1. For a given budget $N$ and beam width $M$, expand the initial set of beams into $K$ independent subtrees.

2. Within each subtree, select the step with the highest score.

3. From the selected nodes, generate $M$ new steps and again retain the one with the highest PRM score in each subtree.

4. Repeat Step (3) until the EOS token is generated or the maximum tree depth is reached.

**Incorporating UAA into DVTS.** We incorporate UAA into DVTS by adapting the pruning and budgeting stages to each subtree. In Step 2, *Uncertainty-Aware Pruning* (UAP) is applied within each subtree to remove high-uncertainty candidates, ensuring that only reliable nodes remain eligible for expansion. In Step 3, *Uncertainty-Aware Budgeting* (UAB) controls the expansion width without enlarging the overall budget: nodes with high uncertainty retain the default allocation of $M$ expansions, while nodes with lower uncertainty have their expansion budgets down-weighted accordingly.

### F.1.4 REWARD BALANCED SEARCH (REBASE)

REBASE (Wu et al., 2024) is a tree search algorithm that balances exploration based on reward scores without expensive rollouts. The procedure is as follows:

1. **Initialization:** Given a question $x$, balance temperature $T_b > 0$, and target number of solutions $N$, sample $N$ initial steps from the policy model to form the first layer of nodes. Set the budget at depth 0 to $B_0 = N$.

2. **Reward assignment:** At depth $i$, assign rewards $R(n_j)$ to each node $n_j$. Count completed solutions $C_i$, and update the budget:

$$B_i \leftarrow B_{i-1} - C_i.$$

If $B_i = 0$, terminate and return the $N$ collected solutions.

3. **Balanced expansion:** For each node $n_j$ at depth $i$, compute its expansion width:

$$W_j = \text{Round}\left( \frac{B_i \exp(R(n_j)/T_b)}{\sum_k \exp(R(n_k)/T_b)} \right).$$

Expand $W_j$ children for node $n_j$ by sampling from the policy model.

4. **Iteration:** Repeat steps (2)–(3) until all $N$ solutions are generated or the maximum tree depth is reached.

**Incorporating UAA into REBASE.** We integrate UAA into REBASE by modifying the reward assignment and expansion stages. In Step 2, *Uncertainty-Aware Pruning* (UAP) is applied to filter out nodes with low uncertainty before computing the reward-based budget. This reduces redundant exploration of unreliable candidates. In Step 3, *Uncertainty-Aware Budgeting* (UAB) adjusts the expansion width $W_j$ by combining reward scores with the uncertainty estimates, so that nodes with lower uncertainty receive fewer budgets.

# G   MAIN RESULTS OF QWEN3-32B

In this part, we evaluate UAA on the *Qwen3-32B* model under the experimental setting described in Section 5, applying it to both Beam Search and MCTS. We set the temperature to 0.7 and adopt other hyperparameters from the original paper (Yang et al., 2025), and run the model in its non-thinking mode. For UAA, we set $\lambda = 0.1$ and other hyperparameters are the same as Section 5.

Table 9 shows that integrating UAA into Beam Search and MCTS consistently improves efficiency while maintaining or slightly increasing accuracy across GSM8K, MATH-500, and AIME24. For Beam Search with the Confidence verifier, UAA reduces token usage and latency by about 30% while yielding up to +7% accuracy gains on the AIME24 dataset. For MCTS, the effect is more dramatic: token consumption and time are reduced by up to 90% (e.g., on MATH-500 with the Confidence verifier), with accuracy remaining stable or improving slightly. Similar trends hold under both Confidence and PRM verifiers, demonstrating that **UAA is a simple plug-in mechanism that substantially lowers computational cost without sacrificing performance.**

Table 5: Main results of Beam Search and MCTS on *Qwen3-32B* across GSM8K, MATH-500, and AIME24. Metrics are reported as mean values.

| Verifier | Method | GSM8K | | | MATH-500 | | | AIME24 | | |
|---|---|---|---|---|---|---|---|---|---|---|
| | | Acc. (%) ↑ | Tokens (k) ↓ | Time (s) ↓ | Acc. (%) ↑ | Tokens (k) ↓ | Time (s) ↓ | Acc. (%) ↑ | Tokens (k) ↓ | Time (s) ↓ |
| Conf. | Beam Search | 98.00 | 3.70 | 81.81 | 86.67 | 10.05 | 282.77 | 33.33 | 33.61 | 997.66 |
| | +UAA | 99.00 | **2.90** | 58.35 | 85.00 | **7.52** | 149.69 | 40.00 | **24.50** | 520.17 |
| | MCTS | 98.00 | 48.45 | 983.83 | 88.00 | 134.47 | 3029.64 | 26.67 | 62.77 | 1322.08 |
| | +UAA | 99.00 | **7.01** | 277.60q | 89.00 | **14.12** | 338.19 | 23.33 | **43.53** | 1061.97 |
| PRM | Beam Search | 97.50 | 3.96 | 109.68 | 86.33 | 10.07 | 305.94 | 33.33 | 35.68 | 1164.62 |
| | +UAA | 98.50 | **2.95** | 60.23 | 86.00 | **7.78** | 161.57 | 36.67 | **27.36** | 640.69 |
| | MCTS | 98.00 | 48.45 | 983.83 | 84.00 | 27.81 | 644.42 | 16.00 | 66.16 | 1537.30 |
| | +UAA | 99.00 | **9.45** | 357.35 | 84.00 | **18.10** | 481.05 | 20.00 | **45.75** | 1210.97 |

# H   MORE EXPERIMENTAL RESULTS

## H.1   RESULTS ON MORE BENCHMARKS

**Results on StrategyQA** In this part, we evaluate the effectiveness of UAA on the logical-reasoning dataset, i.e., StrategyQA (Geva et al., 2021), following the experimental settings in Section 5. Since the PRMs used previously were trained on mathematical data, we adopt confidence values to guide the search process. The results in Table 6 show that UAA consistently enhances both accuracy and efficiency across models and search algorithms. For Beam Search, UAA brings notable accuracy improvements on both *Llama3.1-8B-Instruct* and *Qwen3-32B* while simultaneously reducing token usage and inference time. For MCTS, UAA yields further efficiency gains and lower runtime, while also maintaining accuracy. Overall, these results demonstrate that UAA generalizes well to logical reasoning tasks, providing consistent benefits in both search quality and computational efficiency.

**Results on the Blocksworld dataset** We further evaluate UAA on a realistic planning benchmark, Blocksworld (Valmeekam et al., 2023). Our implementation follows the open-source *LLM Reasoners* library (Hao et al., 2024), which employs confidence-based scoring within Beam Search while using the LLM to generate actions. Experiments are conducted on *Llama3.1-8B-Instruct* and evaluate performance on the two-step easy cases. The beam size and expansion budget of the baseline beam

Table 6: Results of StrategyQA on Beam Search across *Llama3.1-8B-Instruct* and *Qwen3-32B*.

| Method | *Llama3.1-8B-Instruct* | | | *Qwen3-32B* | | |
|---|---|---|---|---|---|---|
| | Acc. (%) ↑ | Tokens ($k$) ↓ | Time ($s$) | Acc. (%) ↑ | Tokens ($k$) ↓ | Time ($s$) |
| Beam Search | 60.67 | 3.23 | 199.97 | 64.33 | 2.02 | 74.54 |
| +UAA | 65.33 | **2.92** | 195.54 | 72.00 | **1.64** | 70.12 |
| MCTS | 69.00 | 6.52 | 257.54 | 75.67 | 3.93 | 120.14 |
| +UAA | 70.33 | 3.91 | 235.81 | 75.33 | **2.31** | 108.99 |

search are set to 4. We set the $\widehat{K} = 2$ and $\widehat{B} = 4$. As reported in Table 7, UAA consistently improves both accuracy and efficiency. For instance, on the *Llama3.1-8B-Instruct* model, UAA reduces token consumption from 1.09k to 0.93k while boosting accuracy from 18.92% to 24.32%. These results demonstrate that UAA generalizes effectively to planning tasks, offering substantial gains in search quality and computational efficiency.

Table 7: Results of Blocksworld on Beam Search across *Llama3.1-8B-Instruct* and *Qwen3-32B*.

| Method | *Llama3.1-8B-Instruct* | | | *Qwen3-32B* | | |
|---|---|---|---|---|---|---|
| | Acc. (%) ↑ | Tokens ($k$) ↓ | Time ($s$) | Acc. (%) ↑ | Tokens ($k$) ↓ | Time ($s$) |
| Beam Search | 18.92 | 1.09 | 130.67 | 62.16 | 0.64 | 1851.54 |
| +UAA | 24.32 | **0.93** | 108.11 | 62.16 | **0.56** | 1645.81 |

## H.2 RESULTS OF GPT-4O

In this section, we further evaluate UAA on GPT-4o. Since model logits and embedding vectors are unavailable in this setting, we employ a PRM to guide the tree-search procedure. Given API budget constraints, we assess the effectiveness of UAA using beam search only. We follow the same experimental setup described in Section 5, and the results are summarized in Table 8. As shown, UAA continues to provide substantial efficiency gains even when applied to a significantly larger and more capable model. Across GSM8K, MATH-500, and AIME24, **UAA consistently reduces token consumption while maintaining accuracy comparable to the baseline.**

These findings indicate that UAA scales effectively to more capable LLMs. Stronger models generally produce more stable and semantically consistent reasoning steps, making UAA's redundancy-aware pruning even more advantageous in such regimes. Although we do not include experiments on proprietary models such as Claude 3 or Gemini 2.5 Pro due to API budget constraints, the GPT-4o results provide strong evidence that UAA remains effective and complementary when deployed with high-end LLMs.

Table 8: Main results of Beam Search on GPT-4o across GSM8K, MATH-500, and AIME24.

| Method | GSM8K | | | MATH-500 | | | AIME24 | | |
|---|---|---|---|---|---|---|---|---|---|
| | Acc. (%) ↑ | Tokens ($k$) ↓ | Time ($s$) ↓ | Acc. (%) ↑ | Tokens ($k$) ↓ | Time ($s$) ↓ | Acc. (%) ↑ | Tokens ($k$) ↓ | Time ($s$) ↓ |
| Beam Search | 94.00 | 17.18 | 58.2 | 68.00 | 37.98 | 242.4 | 13.33 | 71.75 | 712.3 |
| +UAA | 95.00 | **12.96** | 36.5 | 67.00 | **22.76** | 148.7 | 13.33 | **37.50** | 402.8 |

## I DEPTH ANALYSIS OF THE SEARCH TREES

In this section, we present a reasoning-depth analysis for both Beam Search and MCTS, based on the experiments reported in Section 5. As shown in Table 6, integrating UAA into Beam Search and MCTS consistently preserves the effective reasoning depth of the search tree. This demonstrates that UAA reduces token consumption without prematurely truncating the exploration process. Rather than discarding meaningful reasoning steps, UAA removes only redundant expansions while keeping the valid reasoning trajectory fully extendable. Consequently, UAA achieves substantially lower token

usage while maintaining comparable, and occasionally improved, accuracy across GSM8K, MATH-500, and AIME24. These results confirm that **UAA enhances efficiency without compromising the depth or quality of reasoning.**

Table 9: Depth analysis of Beam Search and MCTS on *Llama3.1-8B-Instruct* and *Qwen3-32B* across GSM8K, MATH-500, and AIME24.

| Verifier | Method | *Llama3.1-8B-Instruct* | | | *Qwen3-32B* | | |
|---|---|---|---|---|---|---|---|
| | | **GSM8K** | **MATH-500** | **AIME24** | **GSM8K** | **MATH-500** | **AIME24** |
| Conf. | Beam Search | 5.12 | 9.46 | 17.29 | 13.73 | 23.84 | 38.22 |
| | +UAA | 5.22 | 9.01 | 17.14 | 20.69 | 27.38 | 37.98 |
| | MCTS | 12.45 | 22.75 | 31.74 | 14.25 | 25.46 | 39.15 |
| | +UAA | 15.25 | 20.64 | 32.43 | 15.86 | 25.27 | 39.68 |
| PRM | Beam Search | 4.87 | 8.64 | 14.87 | 15.24 | 23.83 | 38.28 |
| | +UAA | 4.93 | 8.86 | 16.02 | 20.40 | 24.80 | 37.70 |
| | MCTS | 9.51 | 19.37 | 32.91 | 17.25 | 26.5 | 39.94 |
| | +UAA | 9.63 | 16.29 | 35.46 | 18.41 | 25.49 | 38.12 |

## J   ABLATION STUDY ON KEY HYPERPARAMETER

To better understand the influence of the pruning count $(\hat{k})$ and the reduced budget $(\hat{B})$, we conduct a fine-grained ablation study under both confidence-guided and PRM-guided Beam Search (Figures 11 and 12). Across both settings, we report three metrics: accuracy gap, token gap, and time gap relative to the vanilla Beam Search baseline.

Overall, the heatmaps reveal a consistent trend. For accuracy, moderate values, typically near the central region of the search grid, achieve the smallest accuracy drop or even slight improvements. In terms of efficiency (token and time gap), increasing $(\hat{k})$ and decreasing $(\hat{B})$ produce substantial savings, confirming that the proposed uncertainty-aware pruning indeed reduces redundant exploration. This pattern is observed consistently for both confidence and PRM guidance. Importantly, both guidance strategies show that the "sweet spot" lies in a balanced configuration that is aggressive enough to reduce cost, but not too aggressive to harm accuracy.

Taken together, these observations suggest that **the optimal hyperparameter setting typically corresponds to using roughly half of the original pruning and budget values**. In practice, setting $(\hat{k} = k/2)$ and $(\hat{B} = B/2)$ provides a robust trade-off between performance and efficiency across different guidance strategies.

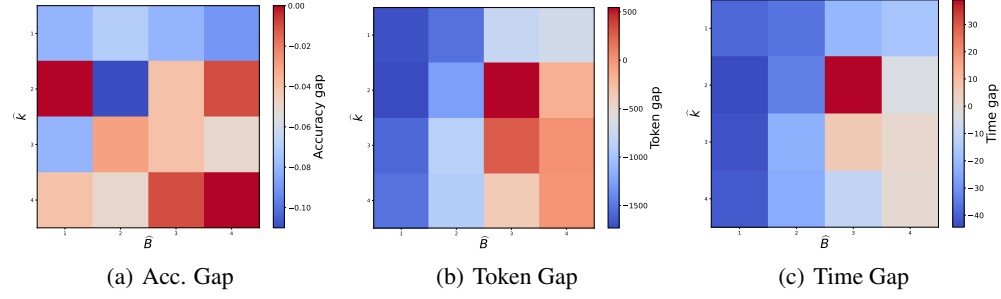

| (a) Acc. Gap | (b) Token Gap | (c) Time Gap |
|---|---|---|

Figure 11: Ablation study on key hyperparameter $(\hat{k}, \hat{B})$ with confidence-guided Beam Search.

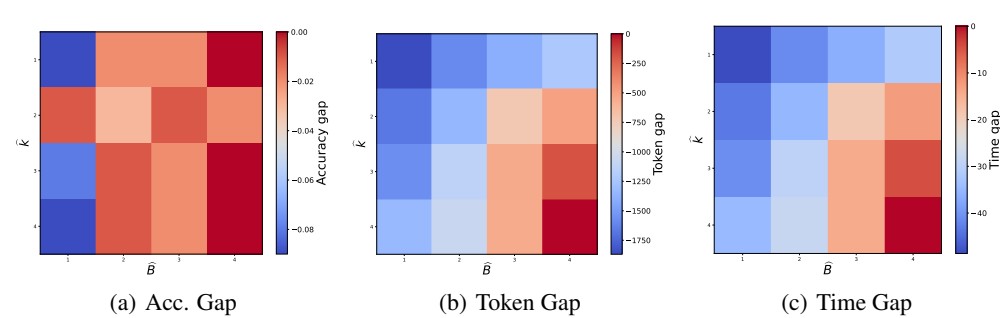

(a) Acc. Gap  (b) Token Gap  (c) Time Gap

Figure 12: Ablation study on key hyperparameter $(\widehat{k}, \widehat{B})$ with PRM-guided Beam Search.

