# OpenReview forum: "Uncertainty-Aware Tree Search for Efficient LLM Reasoning"
_ICLR.cc/2026/Conference — Submitted to ICLR 2026_

### Official Review · Reviewer_UpCm · 2025-10-28

**Soundness:** 3
**Presentation:** 3
**Contribution:** 3
**Rating:** 6
**Confidence:** 4

**Summary:**

The existing tree search methods for multi-step inference in large language models suffer from high computational overhead. The article points out that these methods allocate the same computational budget to all inference branches without considering the uncertainty of the branches themselves. The author found through empirical research that there is a high degree of semantic redundancy in the inference trajectory starting from intermediate steps in tree search inference of large language models, and this redundancy contributes limited to the diversity of final answers. Based on this, the article proposes a plug and play framework for uncertainty aware allocation (UAA) that does not require training, including uncertainty aware pruning (UAP) and uncertainty aware budget (UAB). Experiments show that in Beam Search and MCTS, token consumption and inference time are significantly reduced while maintaining or even improving accuracy.

**Strengths:**

1、The proposed UAA framework is training-free, requires no auxiliary models or fine-tuning, and can be easily integrated into existing tree-search algorithms.
2、The paper provides a clear information-theoretic motivation (Corollary 1) and supports it with empirical evidence, including a pilot study on trajectory redundancy and extensive experiments across multiple models and benchmarks.
3、UAA consistently reduces token usage and inference time across Beam Search, MCTS, DVTS, and REBASE, without compromising accuracy. The result that highlights its generality and robustness.

**Weaknesses:**

1、The experiment is limited to mathematical reasoning tasks. The effectiveness of this framework for other types of reasoning, such as logical reasoning, common sense reasoning, or planning, has not been confirmed, which limits its claimed universality.
2、Key hyperparameters such as the uncertainty threshold λ, pruning count k^, and reduced budget B^ are fixed across experiments. Their sensitivity and adaptability across different tasks or model scales are not systematically analyzed, which may affect reproducibility and deployment in new settings.

**Questions:**

1、How was the intermediate state selected for subsequent redundancy analysis in the pilot experiment of the third section?
2、Have the authors considered evaluating UAA on non mathematical reasoning tasks, such as common sense reasoning ScienceQA and logical reasoning StrategyQA, to better support its universal applicability?
3、Could the authors provide more insight into how the hyperparameters (λ, k^, B^) were chosen? A sensitivity analysis or adaptive strategy for these parameters would strengthen the method's practicality.

---

> ### Author Response · Authors · 2025-11-27
>
> Thank you for your positive and valuable feedback. We address your concerns point by point below:
>
> ### **1. Results on more benchmarks [W1, Q2]**
>
> See General Response §1.
>
> ### **2. Ablation study on key hyperparameter [W2, Q3]**
>
> We thank the reviewer for pointing this out. In the revised version, we provide a more detailed analysis of how these hyperparameters are selected and how sensitive UAA is to them.
>
> **Uncertainty threshold $\lambda$.** We provide an ablation stuyd on $\lambda$ in Figure 3. the result show taht for different $\lambda$,  accuracy  remain flat and close to the base accuracy. Therefore, we recommend the user choose a conservative value $\lambda=0.5$
>
> **Pruning count $\widehat{k}$ and reduced budget $\widehat{B}$.** To better understand the influence of $\widehat{k}$ and $\widehat{B}$, we conduct a fine-grained ablation study under both confidence-guided and PRM-guided Beam Search (See Figures 11 and 12). We report three metrics: accuracy gap, token gap, and time gap relative to the vanilla Beam Search baseline. Overall, the heatmaps show a consistent pattern across both guidance strategies. For accuracy, moderate values, which are typically near the central region of the search grid, incur the smallest accuracy drop and sometimes even yield slight improvements over the baseline. For efficiency (token and time gap), decreasing $\widehat{k}$ and $\widehat{B}$ produces substantial savings, confirming that the proposed uncertainty-aware pruning effectively reduces redundant exploration.
>
> Importantly, both guidance strategies reveal a clear ''sweet spot'': a balanced configuration that is aggressive enough to reduce cost, but not so aggressive that it harms accuracy. Taken together, these observations suggest that **the suggested hyperparameter setting typically corresponds to using roughly half of the original pruning and budget values.** In practice, setting $\widehat{k}=k/2$ and $\widehat{B}=B/2$ provides a robust trade-off between performance and efficiency across different guidance signals.
>
> ### **3. The intermediate state selected for subsequent redundancy analysis [Q1]**
>
>
> Thank you for the thoughtful question. In the pilot study of Section 3, we did not select a particular intermediate state. Instead, our redundancy analysis was conducted over all intermediate states generated during reward-guided beam search.
> Specifically, under a beam width of 20, at each decoding step, every active node in the beam is treated as an intermediate state $s_t$. For each such state, we sample 20 continuations, embed them, and compute pairwise cosine similarities (as described in Appendix B.2). Therefore, the distributions shown in Figure 1 aggregate statistics across all intermediate states, all steps, and all problems in the evaluation subset.
> To avoid ambiguity, we will clarify this explicitly in Section 3 of the revised manuscript.

---

### Official Review · Reviewer_tBWA · 2025-10-29

**Soundness:** 3
**Presentation:** 3
**Contribution:** 3
**Rating:** 4
**Confidence:** 3

**Summary:**

The authors conduct an empirical study to reveal the ubiquitous redundancies among reasoning trajectories in tree-search methods, and demonstrate that such redundancies would ultimately reduce the diversity of the final answers. To address this, this paper presents Uncertainty-Aware Allocation(UAA), a training-free, plug-and-play module that dynamically allocates budgets for each reasoning step in the search tree. UAA is used for both Beam Search and MCTS. Experiments conducted on the GSM8K, MATH500, and AIME24 benchmarks validate that UAA successfully improves the reasoning efficiency of these tree-search algorithms.

**Strengths:**

1. The paper motivation is clear, addressing the critical problem of extensive redundancy within the search space of tree-search algorithms for LLM reasoning. The empirical study reveal that this redundancy is prevalent among reasoning trajectories originating from intermediate steps and compellingly demonstrate that it ultimately diminishes the diversity of the final answers. This insight is a significant contribution to the advancement of tree-search methods for LLM reasoning.

2. UAA improves the reasoning efficiency when it integrates with tree-search algorithms like Beam Search and MCTS on challenging benchmarks, including GSM8K, MATH500, and AIME24.

3. The paper provides theoretical proofs to substantiate its claims.

**Weaknesses:**

1. The experiments is confined to MATH benchmarks (GSM8K, MATH500, AIME24). The method's adaptability and effectiveness across diverse types of datasets remain unverified. This is a notable limitation, particularly as LLM + MCTS methods are increasingly being applied to more realistic and interactive reasoning environments, such as Blocksworld [1] and WebShop [2], where the search dynamics may differ.

2. The empirical study on reasoning trajectory redundancy appears to be limited, as it was conducted using only small-scale LMs and focused exclusively on Beam-Search. And, the paper lacks a corresponding analysis for the MCTS scenario.

[1] Reasoning with language model is planning with world model[J]. At EMNLP.

[2] Webshop: Towards scalable real-world web interaction with grounded language agents[J]. At NeurIPS.

**Questions:**

1. Could the authors provide experiments applying UAA to Beam Search or MCTS on other LLM reasoning tasks (beyond math benchmarks), preferably on more realistic reasoning tasks, such as web-based tasks?

2. Could you provide additional data to demonstrate that the observed reasoning trajectory redundancy also occurs when employing powerful LLMs, such as GPT-4o, within both Beam Search and MCTS? This evidence would be crucial to confirm that the redundancy is an inherent challenge in the search process itself, rather than a byproduct of using less capable models.

3. How does the introduction of UAA impact the final depth of the search trees? Is there a risk that dynamically allocating budgets might lead to insufficient exploration (i.e., pruning promising paths too early) or premature termination of the search?

---

> ### Author Response · Authors · 2025-11-27
>
> Thank you for your positive and valuable feedback. We address your concerns point by point below:
>
> ### **1. Results on more benchmarks [W1,Q1]**
>
> See General Response §1.
>
> ### **2. Empirical studies on large LLM and MCTS [W2]**
>
> Due to the limited API credits, we add experiments on the open-sourced LLM, i.e., Qwen3-32b. More details can be found in General Response §3.
>
> ### **3. Final depth of the search trees [Q3]**
>
> UAA does not reduce the final depth of the search trees. The algorithm only adjusts the local exploration budget of each node based on its uncertainty, but it does not modify the model’s reasoning depth. In other words, UAA adapts the width of the search (by pruning redundant nodes and reducing the budget to low-uncertainty nodes), but it does not change the depth: every branch can still be expanded up to the same depth limit as in the original Beam Search or MCTS.
>
> Empirically, the depth statistics in our additional experiments confirm this behavior. As shown in the table below, adding UAA to Beam Search produces nearly identical depths to the baseline across GSM8K, MATH-500, and AIME24, with only small natural fluctuations. This demonstrates that **UAA preserves the structural depth of the search while improving budget efficiency.** More results can be found in Appendix I of the revised paper.
>
> | Verifier  | Method      | Llama3.1-8B-Instruct |  | |Qwen3-32B|||
> | --------- | ----------- | ------------- | ---------------- | -------------- | ------------ | --------------- | ------------- |
> | |  | GSM8K        | MATH-500            | AIME24         | GSM8K        | MATH-500            | AIME24         |
> | **Conf.** | Beam Search | 5.12          | 9.46             | 17.29          | 13.73        | 23.84           | 38.22         |
> |           | +UAA        | 5.22          | 9.01             | 17.14          | 20.69        | 27.38           | 37.98         |
> | **PRM**   | Beam Search | 4.87          | 8.64             | 14.87          | 15.24        | 23.83           | 38.28         |
> |           | +UAA        | 4.93          | 8.86             | 16.02          | 20.40        | 24.80           | 37.70         |

---

### Official Review · Reviewer_DVC6 · 2025-10-29

**Soundness:** 2
**Presentation:** 3
**Contribution:** 2
**Rating:** 4
**Confidence:** 4

**Summary:**

This paper presents an uncertainty-aware framework denoted as UAA, which is designed to optimize the tree-search reasoning efficiency of Large Language Models (LLMs). Corresponding experiments of the framework are carried out on multiple mathematical reasoning benchmarks.

**Strengths:**

The study’s focus on addressing uncertainty in LLM-based reasoning holds certain merit, as effective uncertainty management is critical for improving multi-step reasoning efficiency.

**Weaknesses:**

1. UAA’s premise that "low-uncertainty steps correspond to semantic redundancy with no contribution to answer diversity" is only validated in math reasoning. This assumption may fail in open-domain tasks
2. Experiments are only conducted on open-source models (Llama3.1-8B-Instruct, Qwen3-32B). For widely used closed-source LLMs (e.g., GPT, Gemini) in industry, the paper only mentions "confidence/PRM variance" as alternative uncertainty proxies but lacks empirical validation, leaving UAA’s practicality unproven.

**Questions:**

1. Is the correlation between uncertainty and "step value" absolute? UAA defaults to the assumption that "low uncertainty equates to low value (thus prunable)". However, in mathematical reasoning, "low-uncertainty yet critical steps" may exist—for instance, "lemma derivation" in proof-based problems, where the step is uniquely determined but a single error would invalidate the entire reasoning process. Why does UAA rely exclusively on uncertainty for pruning, rather than integrating "step correctness scores" to make a holistic judgment?
2. Can experiments on open-domain reasoning tasks (e.g., question answering, dialogue generation, and planning tasks such as WebShop) be incorporated to verify the generalizability of UAA’s core assumption?
3. Will supplementary experiments be conducted on closed-source LLMs (e.g., GPT, Gemini) to validate the effectiveness of alternative uncertainty proxies (e.g., confidence scores/PRM variance) in black-box scenarios?

---

> ### Author Response · Authors · 2025-11-27
>
> Thank you for your positive and valuable feedback. We address your concerns point by point below:
>
> ### **1. Empirical studies on open-domain tasks [W1,Q2]**
> See General Response §3.
>
> ### **2. Results on closed-source LLM [W2,Q3]**
>
> See General Response §2.
>
> ### **3. Explanation of UAA [Q1]**
>
> Thank you for the insightful question. We clarify that UAA does not assume “low uncertainty implies low step value.” In UAA, uncertainty measures the semantic dispersion (diversity) among candidate continuations, rather than the correctness or importance of a step. **Low uncertainty indicates that all candidates are semantically similar, meaning that further exploration of these continuations yields redundant reasoning trajectories without contributing additional diversity.** Therefore, UAA prunes based on redundancy, not value.
>
> This design does not interfere with critical steps. In fact, for proof-based reasoning tasks where the next reasoning step is uniquely determined, semantic redundancy is naturally high, but the valid continuation remains fully preserved. Thus, UAA removes only redundant sibling continuations while keeping the correct trajectory intact, effectively reducing unnecessary token consumption without affecting the validity or progression of the proof.

---

### Official Review · Reviewer_feiC · 2025-10-29

**Soundness:** 3
**Presentation:** 2
**Contribution:** 2
**Rating:** 4
**Confidence:** 3

**Summary:**

The paper introduces a plug-and-play framework named Uncertainty-Aware Allocation (UAA), which solves reasoning computational budgets with the associated uncertainty. First, the paper finds that remaining reasoning trajectories from the same step exhibit high semantic similarity in tree-search methods. Second, it proposes the UAA that accelerates tree-based reasoning via dynamically pruning redundant reasoning steps and allocating expansion budget according to step-wise uncertainty. It is training-free and applicable to both black-box and white-box LLM settings. The results show that UAA consistently reduces inference cost without compromising accuracy and generalizes well across different LLMs settings.

**Strengths:**

1. The paper shows that the ubiquitous semantic redundancy of reasoning trajectories starts from an intermediate reasoning step. Such highly certain reasoning steps will ultimately reduce the diversity of the final answers.
2. The paper proposes Uncertainty-Aware Allocation (UAA), a plug-and-play framework that allocates search budget adaptively according to the step-wise uncertainty with uncertainty-aware pruning (UAP) and uncertainty-aware budgeting (UAB).
3. Empirical evaluations demonstrate that UAA can significantly reduce token consumption and wall-clock time when applied to Beam Search and MCTS.

**Weaknesses:**

1. The current design of UAA adopts fixed thresholds and budgets for compute allocation, which restricts its adaptability across reasoning depths.
2. The proposed method lacks the evaluation of more realistic tasks in the world. The generalization of the method needs further discussion.

**Questions:**

1. How effective is the proposed method on larger and stronger LLMs such as Claude 3 or Gemini 2.5 Pro?
2. How does UAB save costs? As shown in Table 2, tokens only show a significant downward trend when both UAP and UAB are used for tree search. Because "UAP focuses the limited budget on high-value trajectories", why do they together achieve the lowest Tokens?
3. Why do you use two types of value functions to guide search? Because in Tables 1 and 2, it means that the PRM is almost better performance than Conf. for the Acc., Tokens, and Time.

---

> ### Author Response · Authors · 2025-11-27
>
> Thank you for your positive and valuable feedback. We address your concerns point by point below:
>
> ### **1. Thresholds design of UAA [W1]**
>
> Our use of fixed thresholds and budgets is an intentional design choice aimed at simplicity, stability, and reproducibility of UAA. Empirically, our results show that this simple configuration already leads to substantial gains in reasoning efficiency over different baselines. Importantly, **we mitigate depth-related scale differences in uncertainty score by normalizing the uncertainty scores at each depth** (see Appendix F.1.1), so that a single global threshold remains meaningful across different reasoning depths. In this sense, the compute allocation is still state-dependent (via the normalized uncertainty) even though the threshold itself is globally shared.
>
> Moreover, using a single threshold is a common and effective design in prior works [1,2] on adaptive inference, where a global threshold is typically adopted to control computation and accelerate decoding. Our setting follows this widely used and practically successful paradigm, which also keeps the method easy to deploy in new tasks.
>
> Finally, we agree that allowing depth-dependent thresholds and budgets is an interesting extension. Our framework readily supports such variants (e.g., by parameterizing the threshold as a function of depth or other state features), and we consider exploring depth-adaptive scheduling as promising future work that is complementary to the contributions of this paper.
>
> > [1] Fu, Yichao, et al. "Deep think with confidence." arXiv preprint arXiv:2508.15260 (2025).
> >
> > [2] Cao, Lang, et al. "Process Reward Modeling with Entropy-Driven Uncertainty." arXiv preprint arXiv:2503.22233 (2025).
>
> ### **2. Results on more benchmarks [W2]**
>
> See General Response §1.
>
> ### **3. The results of closed-source LLMs [Q1]**
>
> See General Response §2.
>
> ### **4. The functionality of UAB and UAP [Q2]**
>
> UAB regulates the per-node expansion budget, i.e., how many child actions are generated for each candidate node. When a node exhibits low uncertainty, UAB reduces its expansion budget, thereby limiting the number of successor states produced by that node. In contrast, UAP controls which nodes are allowed to be expanded further, effectively reducing the number of parent nodes that participate in the next-level expansion.
>
> Therefore, the two modules operate on orthogonal dimensions:
> - UAP reduces the number of nodes to expand
> - UAB reduces the number of children generated per node
>
> Their combination yields a multiplicative reduction in the total number of expansions. This explains why the lowest token usage occurs only when both UAP and UAB are used together, as observed in Table 2. Although each component individually yields substantial savings, their combination further minimizes the overall computation.
>
> ### **5. Two types of value functions [Q3]**
>
> We include both confidence and PRM because our goal is to demonstrate that UAA is general and works under different value functions. Moreover, confidence is a widely used and readily available signal in tree-search, and it naturally reflects the LLM’s generation confidence, making it a useful indicator for guiding the search process [1,2,3]. Using both value functions verifies that **UAA consistently improves efficiency regardless of the underlying guidance signal.**
>
> > [1] Fu, Yichao, et al. "Deep think with confidence." arXiv preprint arXiv:2508.15260 (2025).
> >
> > [2] Ding, Yifu, et al. "Dynamic parallel tree search for efficient llm reasoning." arXiv preprint arXiv:2502.16235 (2025).
> >
> > [3] Huang, Yuheng, et al. "Look before you leap: An exploratory study of uncertainty measurement for large language models." arXiv preprint arXiv:2307.10236 (2023).

---

### Author Response · Authors · 2025-11-27
**General Response - part 1**

We sincerely thank all reviewers for their thoughtful and constructive feedback. We are grateful for the reviewers' recognition of our contributions: our proposed method is acknowledged as **a training-free, plug-and-play and effective framework** [feiC, DVC6, tBWA, UpCm], which is supported by **a clear information-theoretic motivation** [tBWA, UpCm]. Moreover, this work **addresses uncertainty in LLM-based reasoning**, which holds substantial **merit** [DVC6], and offers **important insight** into the search space structure of LLM tree search [feiC, tBWA]. Extensive experiments further demonstrate the **effectiveness** of UAA: it can significantly reduce both token consumption and wall-clock time [feiC], and it **generalizes well** across different LLMs and search settings [feiC, tBWA, UpCm].

In the following, we respond to the reviewers’ comments and concerns point by point. We appreciate the reviewers’ feedback, which has helped us further strengthen the manuscript. **The main changes are summarized below**:

1. Added empirical motivation experiments on an additional dataset (StrategyQA), search algorithm (MCTS), and large LLM (Qwen3-32B) in **Appendix C**.  [DVC6, tBWA]
2. Added results on an open-domain task (StrategyQA) and a realistic planning benchmark (Blocksworld) in **Appendix H.1**. [feiC, tBWA, UpCm]
3. Added results of UAA on a closed-source model, GPT-4o, in **Appendix H.2**. [feiC, DVC6]
4. Added a depth analysis of UAA in **Appendix I**. [tBWA]
5. Added key parameter analysis experiments in **Appendix J**. [UpCm]

For clarity, all updated content is highlighted in **blue** in the revised paper. We believe these clarifications and additional experiments strengthen both the understanding and practical applicability of our work.

Below, **we address three recurring concerns: requests for additional benchmark results, results on a closed-source LLM, and further empirical studies, by providing the corresponding experimental evidence.** We also provide additional details for reviewer-specific questions in the individual responses.

---

### **1. Results on more benchmarks**

We sincerely thank reviewers for their thoughtful suggestions. To evaluate the generalizability  of our UAA, we conduct additional experiments on a common sense logical reasoning dataset (StrategyQA [1]), and a realistic planning dataset (Blocksworld [2]). The results are presented as follows.

**Results of StrategyQA** We conduct the Beam search and MCTS for both Llama3.1-8B-Instruct and Qwen3-32B. Since the PRMs previously were trained on mathematical data, we adopt confidence values to guide the search process. The results of confidence-guided beam search are presented in the following table. It can be seen that **UAA consistently improves reasoning efficiency while preserving answer accuracy**. Particularly, UAA reduces token usage by over 20% for both Llama3.1-8B-Instruct and Qwen3-32B, respectively. More results for MCTS can be found in Appendix H.1.

| Method        | Acc. (%) ↑ | Tokens (k) ↓ | Time (s) | Acc. (%) ↑ | Tokens (k) ↓ | Time (s) ↓|
|---|---|---|---|---|---|---|
|| **Llama3.1-8B-Instruct** | | | **Qwen3-32B** | | |
| Beam Search   | 60.67  | 3.23  | 199.97   | 64.33| 2.02 | 74.54    |
| +UAA  | **65.33** | **2.92**| 195.54| **72.00**  | **1.64**| 70.12  |


**Results of Blocksworld** Due to limited computational resources, we evaluate UAA on the Blocksworld dataset using Beam Search with Llama3.1-8B-Instruct and Qwen3-32B. Our implementation follows the open-source LLM Reasoners library [3], which integrates confidence-based Beam Search relying on the LLM to generate actions. As shown in the Table below, UAA consistently enhances both accuracy and efficiency. For example, on the Llama3.1-8B-Instruct model, UAA reduces token consumption from 1.09k to 0.93k while improving accuracy from 18.92% to 24.32%.

| Method | Model | Acc. (%) ↑ | Tokens (k) ↓ | Time (s) ↓|
| ----------- | -------------------- | ---------- | ------------ | -------- |
| Beam Search | Llama3.1-8B-Instruct | 18.92 | 1.09 | 130.67 |
| +UAA | Llama3.1-8B-Instruct | 24.32 | **0.93** | 108.11 |
| Beam Search | Qwen3-32B | 62.16 | 0.64 | 1851.54 |
| +UAA | Qwen3-32B | 62.16 | **0.56** | 1645.81 |


These results demonstrate that **UAA generalizes effectively to common sense reasoning datasets and planning tasks, offering substantial gains in computational efficiency**.

Reference

> [1] Geva, Mor, et al. "Did aristotle use a laptop? a question answering benchmark with implicit reasoning strategies." Transactions of the Association for Computational Linguistics 9 (2021): 346-361.
>
> [2] Valmeekam, Karthik, et al. "On the planning abilities of large language models (a critical investigation with a proposed benchmark)." arXiv preprint arXiv:2302.06706 (2023).
>
> [3] Hao, Shibo, et al. "Llm reasoners: New evaluation, library, and analysis of step-by-step reasoning with large language models." arXiv preprint arXiv:2404.05221 (2024).

---

> ### Author Response · Authors · 2025-11-27
> **General Response - part 2**
>
> ### **2. The results of Closed-source LLM**
>
> To assess the applicability of UAA on more powerful, closed-source LLMs, we additionally evaluate our method on GPT-4o. Due to limited API credits, we do not evaluate proprietary models such as Claude 3 or Gemini 2.5 Pro. Since logits and embeddings are unavailable in this setting, we instead adopt a PRM, i.e., Llama3.1-8B-PRM-Deepseek-Data [1], to guide tree search under the same experimental configuration as Section 5. As shown in the table below, UAA continues to deliver substantial efficiency gains across various benchmarks like GSM8K, MATH-500, and AIME24. Notably, it consistently reduces token consumption while maintaining accuracy comparable to the baseline. These results demonstrate that UAA is also applicable to a wide range of closed-source LLMs. We have added these results to the new Appendix H.2.
>
>  Method      | GSM8K   ||  | MATH500  | |   | AIME24|   |   |
> | -------- | ----------- | --------------- | ------------------ | ---------------- | ----------------- | -------------------- | ------------------ | ---------------- | ------------------- |
> |  | Acc (%)           | Tokens (k)              | Time (s)           | Acc (%)           | Tokens (k)              | Time (s)    |Acc (%)           | Tokens (k)              | Time (s)             |
> | Beam Search | 94.00           | 17.18              | 58.2             | 68.00             | 37.98                | 242.4              | 13.33            | 71.75               | 712.3             |
> | **+UAA**    | 95.00       | **12.96**          | **36.5**         | 67.00         | **22.76**            | **148.7**          | 13.33        | **37.50**           | **402.8**         |
>
>
>
> Reference
>
> > [1] Dong, Hanze, et al. "Rlhf workflow: From reward modeling to online rlhf." arXiv preprint arXiv:2405.07863 (2024).
>
> ### **3. More empirical motivation studies**
>
> Thanks for the positive and valuable feedback. In the revision, we have added additional empirical studies covering MCTS, a larger LLM (Qwen3-32B), and an open-domain question answering benchmark (StrategyQA), addressing the reviewer’s request to test UAA’s assumption beyond math reasoning and Beam Search. For the analysis of reasoning trajectory similarity and the Spearman correlation between trajectory similarity and the diversity of final answers, the detailed figures are provided in Appendix C. The mean statistics reported below reveal a consistent trend across search algorithms (Beam Search and MCTS), model scales (8B–32B), and task types (math and QA): LLM-generated reasoning trajectories exhibit substantial redundancy, and exploring a highly certain reasoning step, whose continuations are semantically similar, contributes little to the diversity of final answers. Therefore, **these results verify the generalizability of UAA’s core assumption across different model scales, search algorithms, and task domains, which also explains why UAA remains effective across such diverse settings.**
>
> | Method        | Model | Dataset | Mean Similarity | Mean Spearman Correlation |
> |---------------|-------------|---------------|----------|--------|
> | MCTS| Llama3.1-8B-Instruct |GSM8K | 0.91 | -0.37 |
> | MCTS  | Qwen3-32B       |  GSM8K  |  0.92 |-0.52 |
> | Beam Search   | Qwen3-32B       |  GSM8K  | 0.96 | -0.66|
> | Beam Search   | Llama3.1-8B-Instruct      |  StrategyQA  | 0.95|  -0.41 |
> | Beam Search   | Qwen3-32B       |  StrategyQA  | 0.96  | -0.37|

---

### Author Response · Authors · 2025-12-02
**Summary of Rebuttal**

All four reviewers expressed overall positive views on our research problem, motivation, and empirical effectiveness. Their concerns fall into three categories, all of which are fully addressed in our rebuttal with new analyses, clarifications, and extensive additional experiments.

**1. Effectiveness on more benchmarks**

  Reviewers [feiC, tBWA, UpCm] questioned whether the proposed method remains effective on additional benchmarks.

  **Our response**: We added results on StrategyQA and Blocksworld with Llama3.1-8B-Instruct and Qwen3-32B, demonstrating that UAA generalizes well to both commonsense reasoning and realistic planning tasks, while delivering substantial computational savings and maintaining accuracy comparable to the baseline.

**2. Performance on closed-source LLMs**

  Reviewers [feiC, DVC6] raised concerns about the effectiveness of UAA on closed-source models.

  **Our response**: We conducted new experiments applying UAA to Beam Search with GPT-4o on GSM8K, MATH-500, and AIME24. The results show that UAA consistently reduces token consumption while maintaining accuracy comparable to the Beam Search baselines.

**3. Additional empirical motivation experiments**

  Reviewers [DVC6, tBWA] requested further motivation analyses on open-domain tasks, larger LLMs, and MCTS.

  **Our response**: We supplemented our motivation analysis with (i) an additional open-domain dataset (StrategyQA), (ii) another search algorithm (MCTS), and (iii) a larger LLM (Qwen3-32B) (see Appendix C). These results corroborate that LLM-generated reasoning trajectories exhibit substantial redundancy and that exploring highly certain reasoning steps, whose continuations are semantically similar, contributes little to the diversity of final answers.

**4. Depth analyses of the searching trees**

  Reviewer tBWA is concerned that our method may reduce the search depth of the reasoning trees.

  **Our response**: We supplemented depth analysis under confidence-guided and PRM-guided Beam Search, showing that UAA preserves the depth of the reasoning trees while substantially improving budget efficiency.

**5. Ablation study on key hyperparameters**

  Reviewer UpCm requested more sensitivity analysis for the hyperparameters.

  **Our response**: Building on our original experiments on the uncertainty threshold, we further conduct a detailed ablation study on the pruning count and reduced budget. The results show that (i) UAA is robust to the uncertainty threshold, and (ii) for both confidence-guided and PRM-guided Beam Search, there exists a clear sweet spot at around half of the original pruning and budget values, which maintains accuracy while significantly reducing token and time cost.

**Overall assessment**

  We believe we have comprehensively addressed all concerns raised by the reviewers through new experiments, deeper analysis, and clearer explanations. The added results further validate the generality, practicality, and empirical robustness of UAA across models, datasets, and search algorithms.

---

### Meta-Review · Area_Chair_Ernc · 2025-12-19

**Summary:**

Tree search is an important method for LLM reasoning. This paper presents an uncertainty-aware framework denoted as UAA for tree search. The authors show that their method is promising.

The strengths of the paper are described as follows:
- A training-free and plug-and-play framework that is easy to integrate.
- Clear information-theoretic motivation.
- Empirically effective on the evaluated tasks

However, there are also some key weaknesses that were pointed out by the reviewers:

- Limited novelty:  the core idea does not clearly differentiate itself from prior uncertainty-aware or confidence-based methods.
- Insufficient experimental and analytical support for the design choices of UAA.
- Lack of comparisons with more recent related methods.
- Missing empirical studies on open-domain tasks and closed-source LLMs in the original submission (partially addressed during rebuttal).

**Reviewer Concerns:**

Reviewers questioned whether the proposed method remains effective on additional benchmarks, its performance on closed-source LLMs, and the sensitivity to hyperparameters. The concerns regarding additional benchmarks, closed-source LLMs, and hyperparameter sensitivity were partially addressed in the rebuttal. However, the lack of insight into the core design of the method and the limited originality remain unresolved.

**Reviewer Scores:**

Based on the reviews and the current state of the paper, it is not ready for acceptance. I recommend that the authors address the above concerns and strengthen both the methodological justification and empirical evaluation before resubmission. The overall scores are expected to remain unchanged (4, 4, 4, 6).

---

### Decision · Program_Chairs · 2026-01-26

Reject